# The hypolipidemic effect of MI-883, the combined CAR agonist/ PXR antagonist, in diet-induced hypercholesterolemia model

Jan Dusek[1,16], Ivana Mejdrová[2,16], Klára Dohnalová[3,4], Tomas Smutny[1], Karel Chalupsky[3], Maria Krutakova[1], Josef Skoda[1], Azam Rashidian[5], Ivona Pavkova[6], Kryštof Škach [2], Jana Hricová [2], Michaela Chocholouskova[7], Lucie Smutna [1], Rajamanikkam Kamaraj [1], Miloš Hroch [8], Martin Leníček [9], Stanislav Mičuda[10], Dirk Pijnenburg[11], Rinie van Beuningen[11], Michal Holčapek [7], Libor Vítek[9,12], Magnus Ingelman-Sundberg [13], Oliver Burk [14], Thales Kronenberger [5,15], Radim Nencka [2]✉ & Petr Pavek [1]✉

Constitutive androstane receptor (CAR) and pregnane X receptor (PXR) are closely related nuclear receptors with overlapping regulatory functions in xenobiotic clearance but distinct roles in endobiotic metabolism. Car activation has been demonstrated to ameliorate hypercholesterolemia by regulating cholesterol metabolism and bile acid elimination, whereas PXR activation is associated with hypercholesterolemia and liver steatosis. Here we show a human CAR agonist/PXR antagonist, MI-883, which effectively regulates genes related to xenobiotic metabolism and cholesterol/bile acid homeostasis by leveraging CAR and PXR interactions in gene regulation. Through comprehensive analyses utilizing lipidomics, bile acid metabolomics, and transcriptomics in humanized PXR-CAR-CYP3A4/3A7 mice fed high-fat and high-cholesterol diets, we demonstrate that MI-883 significantly reduces plasma cholesterol levels and enhances fecal bile acid excretion. This work paves the way for the development of ligands targeting multiple xenobiotic nuclear receptors. Such ligands hold the potential for precise modulation of liver metabolism, offering new therapeutic strategies for metabolic disorders.

The pregnane X receptor (PXR) and constitutive androstane receptor (CAR) are related xenobiotic-sensitive nuclear receptors belonging to the NR1I group. They were originally characterized as regulators of xenobiotic clearance[1,2]. Later, their critical roles in various liver functions, including bile acid, cholesterol, bilirubin, lipid, and glucose metabolism and disposition, emerged[1,3,4]. The human CAR exhibits unique properties, including high constitutive activity and the ability for both direct and indirect activation[4,5]. In contrast, PXR is activated mainly in a ligand-dependent manner with numerous diverse structural ligands[4]. Both CAR and PXR have been suggested to regulate cholesterol and bile acid homeostasis; however, recent compelling data indicate that CAR and PXR ligands have opposing effects on plasma cholesterol levels under nutritional stress[3,6,7].

There is ample evidence for a significant impact of murine Car activation on hypercholesterolemia and cholesterol homeostasis in various animal models of hypercholesterolemia; however, clinical data are lacking, as there is no known CAR ligand for human therapy. In mice, Car activation by TCPOBOP, the prototype murine Car ligand, has been shown to prevent elevated liver and serum cholesterol concentrations caused by a diet containing 1% cholesterol, due to increased bile acid synthesis, metabolism, and excretion, and upregulated removal of low-density lipoprotein (LDL)[8]. In another study,

the authors showed that Car activation promoted the final step of reverse cholesterol transport (transport of $^3$H-cholesteryl esters from HDL to feces) by 50% and reduced the total and hepatic cholesterol accumulation in hypercholesterolemic genetic *Ldlr*$^{-/-}$ and *ApoE*$^{-/-}$ mouse models[9]. Mechanistically, Car activation with TCPOBOP increased fecal excretion of $^3$H-labeled bile acids, mainly muricholic acid (MCA) and deoxycholic acid (DCA), by ~40%, but did not stimulate fecal excretion of neutral sterols. This was mainly due to the increased synthesis of bile acids (BAs), sulfate conjugation, and BA excretion into bile, but decreased BA reabsorption due to the suppression of the ileal apical sodium-dependent bile acid transporter (Asbt) and decreased cholesterol excretion into bile via downregulation of Abcg5/8 transporters[9]. In parallel, TCPOBOP decreased cholesterol content both in the liver (by 25%) and in the whole body by nearly 50% after 2 months[9], and suppressed atherosclerotic lesions in the aortic valves in TCPOBOP-treated *Ldlr*$^{-/-}$ mice by 60% ($P < 0.001$)[10].

Recent systematic clinical studies provide convincing mechanistic evidence for PXR-induced hypercholesterolemia in humans[6,11]. PXR activation elevates plasma LDL cholesterol (LDL-C) levels, triggers the nuclear accumulation of active SREBP2 protein (a master regulator of the cholesterol synthesis), and activates the SREBP2-INSIG1-HMGCR axis. It also increases circulating proprotein convertase subtilisin/kexin type 9 (PCSK9) protein and Niemann-Pick C1-like 1 (NPC1L1)/Npc1l1 expression in the intestine in PXR$^{fl/fl}$ mice, stimulating cholesterol synthesis[6,11,12]. Consistently, genetic association studies support the link between *NR1I2* gene polymorphisms and plasma LDL-C levels[13].

Animal studies have shown that Pxr ablation leads to a significant reduction in plasma cholesterol and atherosclerosis[14,15]. In animal genetic hypercholesterolemic/proatherogenic models, PCN (pregnenolone 16α-carbonitrile) has been found to elevate total cholesterol levels and accelerate atherosclerosis in *ApoE*$^{-/-}$ mice[16]. It also increases plasma total cholesterol and VLDL concentrations while decreasing HDL levels in both ApoE3-Leiden (E3L) and ApoE3-Leiden.CETP mice[17]. Furthermore, certain animal studies suggest that PXR activation with quetiapine and efavirenz can induce hypercholesterolemia and liver steatosis through upregulation of the squalene epoxidase gene (*Sqle*)[12,18]. Collectively, these results suggest that parallel activation of the CAR receptor, combined with the inhibition of PXR, could be a promising therapeutic intervention for hypercholesterolemia.

In this study, we introduce MI-883, a compound exhibiting notable PXR antagonist/inverse agonist activities, derived from our library of novel human CAR agonists[19]. This discovery led us to investigate whether MI-883, acting as a CAR agonist while antagonizing PXR, could elicit combined effects on cholesterol and bile acid homeostasis. Our proof-of-concept studies were conducted in humanized PXR-CAR-CYP3A4/3A7 mice with diet-induced hypercholesterolemia.

We demonstrate the specificity, stability, safety, and favorable pharmacokinetic properties of MI-883. Extensive characterization in humanized PXR-CAR-CYP3A4/3A7 mice revealed a significant reduction in plasma cholesterol levels following treatment with MI-883 in animals subjected to high-fat or high-cholesterol diets. Furthermore, employing mechanistic studies in primary human hepatocytes and PXR/CAR-ablated HepaRG cells, along with BA metabolomics, transcriptomic, and biochemistry analyses, we elucidated the detailed impact of MI-883 on cholesterol homeostasis.

Thus, this study demonstrates the translational development of the first potent combined agonist/antagonist targeting nuclear receptors within the NR1I group and highlights its potential application in the treatment of hypercholesterolemia.

## Results
### MI-883 as a dual CAR/PXR ligand
The compound MI-883 (2-chloro-5-((4-(2-(4-chlorophenyl)-7-fluoroimidazo[1,2-*a*]pyridin-3-yl)-1*H*-1,2,3-triazol-1-yl)methyl)benzamide) (Fig. 1a) is a fluorinated derivative of compound 39 recently proposed

by our team[19]. The specific fluorination protects the pyridine ring of the compound from hydroxylation and significantly extends the metabolic stability and biological half-life of MI-883 in comparison with compound 39[19] and the prototype CAR agonist CITCO (6-(4-chlorophenyl)imidazo[2,1-*b*][1,3]thiazole-5-carbaldehyde-*O*-(3,4-dichlorobenzyl)oxime) in human and mouse liver S9 fractions (Supplementary Fig. 1, Table S1). Importantly, the fluorination does not interfere with the activation of the human CAR receptor.

MI-883 displayed high potency (EC$_{50}$ = 73 ± 20 nM) in TR-FRET CAR coactivator assay with recombinant CAR ligand binding domain (LBD) (Fig. 1b). It significantly stimulates CAR LBD assembly (Fig. 1c), CAR3 variant activation (Fig. 1d), and the translocation of EGFR-CAR+A hybrid protein into nucleus, which is an initial step of CAR activation (Fig. 1e). MI-883 significantly upregulated *CYP2B6* mRNA in HepaRG and primary human hepatocytes (PHH), but not in HepaRG KO CAR cells with ablated CAR expression (Fig. 1f, g).

In TR-FRET PXR LBD Competitive binding assay, MI-883 competes with fluorophore binding to recombinant PXR LBD with IC50 = 0.10 ± 0.02 μM (Fig. 1h). In transient transfection luciferase reporter assays, MI-883 significantly suppressed both basal (IC$_{50}$ = 2.03 ± 0.3 μM) and rifampicin-induced PXR activation (IC$_{50}$ = 3.60 ± 0.4 μM) in HepG2 cells (Fig. 1i) or PXR-overexpressing H-P cells[20] (Supplementary Fig. 2). The failure of MI-883 to inhibit the high constitutive activity of the ligand binding pocket (LBP)-obstructed mutant of PXR (mutant PXR LBP) suggests that MI-883 binds directly to the LBP, which is further supported by the stimulation of the PXR-LBD assembly. Altogether, these data suggest that MI-883 exerts PXR inhibition through interaction with LBP (Fig. 1j, k).

Additionally, treatment of cells with MI-883 results in significant *CYP3A4* mRNA downregulation (*P* value 0.0302 for 10 μM, CI 0.05282 to 0.8072, ANOVA with Dunnett's test), pER6-luc luciferase construct inhibition below the control (basal) expression (*P* value < 0.0001, ANOVA with Dunnett's test), and significant suppression of PXR ligand activity (ANOVA with Tukey's test)(Fig. 1l, m). MI-883 has very similar activity in inhibiting PXR as the SPA70 PXR antagonist in the assays[21].

### Pharmacokinetic properties of MI-883 and preclinical toxicity testing
In the pharmacokinetic study involving C57BL/6N mice, MI-883 demonstrated a biological half-life of 217 min following intraperitoneal application and 170 min following peroral application at a dosage of 10 mg/kg (Supplementary Fig. 3, Table S2). MI-883 is significantly distributed into the liver, but not the brain and it has been excreted mainly via bile (Supplementary Fig. 3b, c). Finally, we found that MI-883 is a weak inhibitor of CYP3A4 activity but does not suppress CYP2B6 activity (Supplementary Fig. 3d, e). We observed no other interactions with other tested nuclear receptors in reporter gene assays with MI-883 (Supplementary Fig. 4).

MI-883 is not genotoxic in the reverse-mutation *Salmonella typhimurium* Ames test (Supplementary, Chapter 17, 20). In the NOAEL/Repeated dose 28-day oral toxicity study in Rats (OECD Guideline No. 407) we observed no toxicity of the compound up to 10 mg/kg (Supplementary, Chapter 18).

### MI-883 stabilizes wild-type CAR but displays distinct PXR binding modes
Molecular dynamics (MD) simulations propose a U-shaped conformation of MI-883 within the wtCAR-LBD, similar to CITCO (Fig. 2a, b), which is stabilized by a set of hydrophobic (residues F161 and Y224, ~100% of the analyzed trajectory), π-mediated (H203 imidazole ring, ~85-100%, Fig. 2c), and the polar interactions (H203, ~50-60%, Fig. 2d), shared by both ligands. CITCO and MI-883 also share similar agonistic structural features, such as the calculated distance between H12 and H3 (~12 Å), as well as hydrogen bonds between K195 and S348 (Fig. 2e, f).

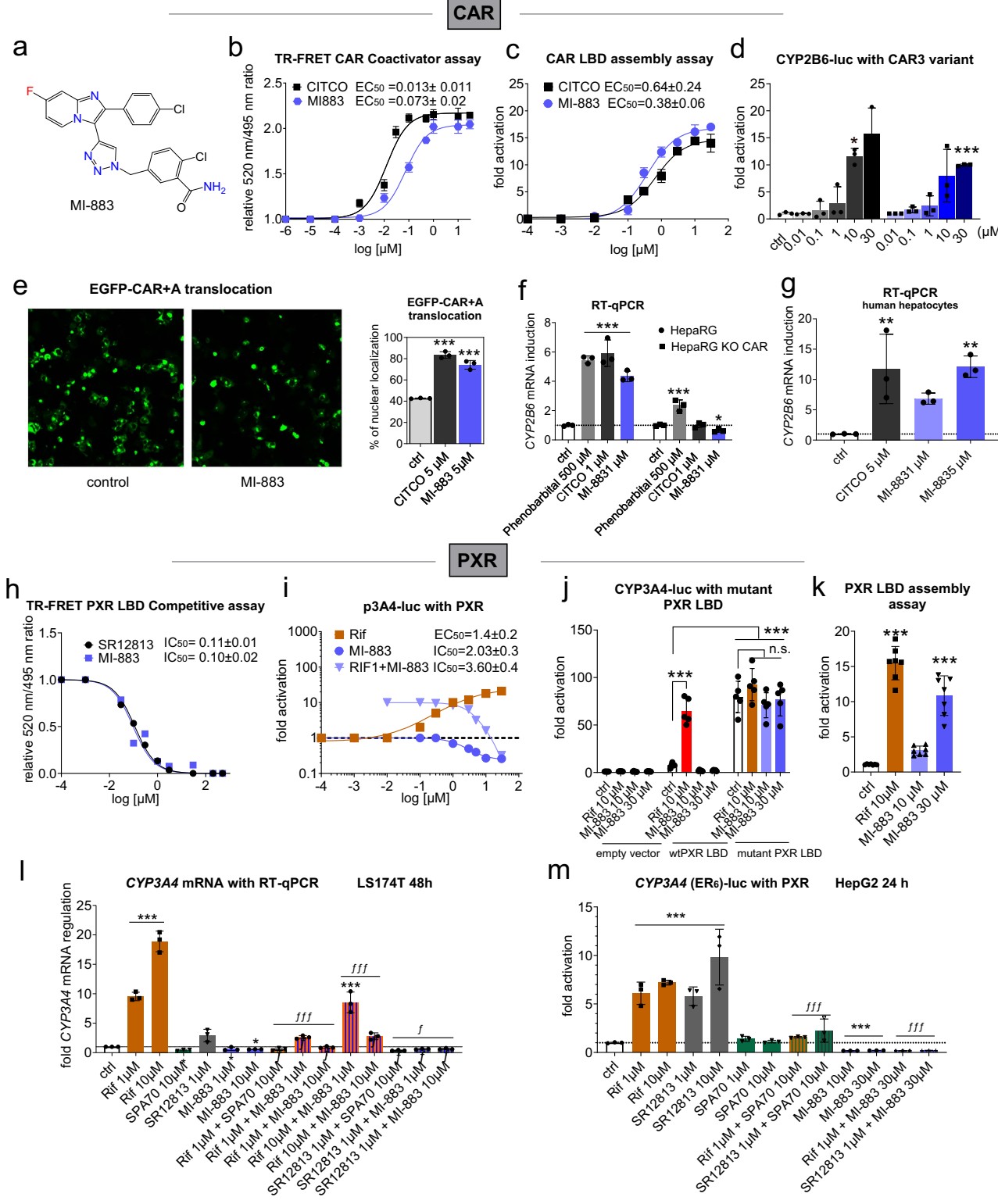

The suggested binding mode of MI-883 in PXR-LBD (Fig. 2g, h), in comparison to SR12813, is supported by the orientation of its 7-fluoroimidazo[1,2-a]pyridine moiety into the hydrophobic sub-pocket. MI-883 is stabilized there through hydrophobic interactions with W299 and Y306 (~40%), while this hydrophobic interaction rises to ~70% with F288. We further observed that only 24% of direct H-bond interactions involve H407 (Fig. 2j), the critical residue for ligand binding[22,23]. Overall, both ligands are stabilized by the highly similar hydrophobic interaction profile, whereas MI-883 shows

additional polar interactions (*see* Supplementary, Chapter 8.1.). In NRs, αAF-2 (part of H3, H4, and H12) forms a suitable platform for the coregulators binding on the LBD surface[24]. Surprisingly, in the presence of MI-883, the distance between H12 and H3 is slightly decreased (median of 10.3 Å) compared to SR12813 (median values ~11 Å) (Fig. 2k). Our result also shows that the distance between C207 (on C-terminus of H2′) and A312 (on β4-H6 loop) forming a water channel[25] is smaller with MI-883 (12.1 Å) than that of PXR/SR12813 (18.6 Å) (Fig. 2l). An earlier study[23] suggested that the closeness of

**Fig. 1 | MI-883 is a CAR agonist and PXR antagonist/inverse agonist. a** Formula of compound MI-883. **b** MI-883 interacts with the recombinant human CAR LBD in the TR-FRET CAR coactivation assay ($n = 3$). **c** MI-883 activates human CAR in the CAR LBD assembly assay ($n = 3$). **d** MI-883 activates the CAR3 variant in the luciferase gene reporter assay with an expression vector for human CAR variant 3 (XM_005245697.4) in HepG2 cells treated with CITCO or MI-883 for 24 h ($n = 3$). **e** MI-883 stimulates EGFP-CAR+A fusion protein cytoplasm-nucleus translocation in COS-1 cells (magnification 40×)($n = 3$). **f** MI-883 significantly induces *CYP2B6* mRNA in HepaRG and **g** primary human hepatocytes, but not in HepaRG KO CAR cells as determined using RT-qPCR expression analysis. **h** MI-883 inhibits the binding of a fluorescent probe to the PXR LBD in TR-FRET PXR LBD Competitive assay. **i** MI-883 inhibits activation of *CYP3A4* gene promoter-based luciferase construct (CYP3A4-luc) in the absence and presence of rifampicin in luciferase reporter gene assay performed in HepG2 cells treated with rifampicin (Rif) and MI-883 for 24 h ($n = 3$). **j** MI-883 does not activate PXR LBP with mutations (S208W/S247W/C284W, mutant PXR LBD) in luciferase reporter gene assay with a *CYP3A4* gene promoter-based luciferase construct in HepG2 cells treated with rifampicin (RIF) and MI-883 for 24 h ($n = 5$). ***$P$ value < 0.001; statistically significant effects *vs.* wtPXR LBD control (ctrl) sample. *n.s.* not statistically significant. **k** MI-883 stimulates PXR LBD assembly with a fusion protein of GAL4 and helix 1 (residues 132–188) and a fusion protein of VP16 and helix 2-12 part (residues 189-434 of PXR-LBD) in HepG2 cells ($n = 7$). **l** MI-883 suppresses rifampicin- and SR12813-mediated *CYP3A4* mRNA induction and basal *CYP3A4* mRNA expression in LS174T cells. **m** The PXR-responsive CYP3A4(pER6)-luc luciferase reporter construct is inhibited by MI-883 in the presence or absence of PXR ligands rifampicin (Rif) and SR12813 in transiently transfected HepG2 cells treated for 24 h. *$^f$$P$ value < 0·05; ** $P$ < 0.01; ***$^{fff}$ $P$ < 0.001; statistically significant effects *vs.* control (ctrl, *) or *vs.* PXR ligand-mediated effect ($^f$) (ANOVA with multiple comparisons). All experiments are biological triplicates ($n \geq 3$) each done in technical triplicates. Data represent means ± S.D.

this channel (smaller distances) could be a characteristic for antagonism binding (Supplementary, Chapter 8.2).

## Interactions of CAR and PXR with coactivators/corepressors by MI-883

MARCoNI nuclear receptor–coregulator interaction profiling revealed that MI-883 attracts several coactivators, including NCOA1, NCOA2, and NCOA3 proteins, in a manner similar to CITCO (Fig. 2m). In contrast, clotrimazole, a CAR inverse agonist, exhibits a nearly opposite pattern by releasing most of the coactivator proteins (see MARCoNI dataset in Supplementary Fig. 9). In cellular experiments with various two-hybrid assays, we confirmed that MI-883 did not recruit nuclear receptor coactivators NCOA1, NCOA2, and MED1 to PXR, but instead disrupted the interaction with the corepressor NCOR2. Additionally, we showed that MI-883 blocked the recruitment of coactivator NCOA1 by agonist rifampicin (Fig. 2n).

## CAR/PXR crosstalk in target gene regulation by MI-883

Next, we analyzed the influence of MI-883 on the dually regulated target genes *CYP3A4* and *CYP2B6* by the closely related CAR/PXR nuclear receptors (Fig. 3a). In differentiated HepaRG, HepaRG KO PXR or HepaRG KO CAR cells, we found that MI-883 suppresses rifampicin-induced and basal *CYP3A4* mRNA expression in a PXR-dependent manner (Fig. 3b). We observed a very similar pattern of *CYP3A4* mRNA regulation in PHH (Fig. 3c). However, we noted more complex regulation for *CYP2B6* mRNA in HepaRG (Fig. 3b) and PHH isolated from two donors (Fig. 3c), suggesting that CAR activation dominates PXR antagonistic activity regulation (Fig. 3b, c). SPA70, a prototype PXR antagonist/inverse agonist, suppressed both rifampicin-induced (antagonist) and basal *CYP3A4* and *CYP2B6* mRNA expression (inverse agonist) similarly to MI-883 in HepaRG cells (Supplementary Fig. 10). Thus, MI-883 exhibits both PXR antagonistic and inverse agonistic activities in regulating these genes. In 3D primary human hepatocyte spheroids from three donors (HJK, AKB, and JEL), MI-883 systematically upregulated *CYP2B6* mRNA but significantly downregulated *CYP3A4* mRNA expression (Fig. 3d). In the livers of hPXR/hCAR/CYP3A mice, we found no significant *CYP3A4* mRNA upregulation after three intraperitoneal (*i.p.*) applications of MI-883 (10 mg/kg) and no significant stimulation of *CYP3A4* mRNA induction by rifampicin ($P = 0.051$, two-way ANOVA, $P = 0.0002$) (Fig. 3e). These results suggest that MI-883 displays the dominant PXR antagonistic effect on *CYP3A4* expression (Fig. 3b–d), while demonstrating a CAR dominant effect on *CYP2B6*/*Cyp2b10* mRNA induction (Fig. 3b–d).

We also examined if the key genes involved in cholesterol, bile acid synthesis, and lipid regulation are coregulated by MI-883 in HepaRG, HepaRG KO CAR, or HepaRG KO PXR cells. Since the Srebp2 protein, a master regulator of cholesterol homeostasis genes, is reported to be controlled by PXR, we used 24-h treatment intervals to monitor direct PXR-mediated effects. We mainly focused on target genes of PXR regulated by human or mouse ligands (such as *SQLE/Sqle*, *Cyp7A1*, *Srebf1*, *Pcsk9*, *Ldlr*, *Hmgcs2* and *Hmgcr* mRNA, Srebp2 protein but not *mRNA*)[11,26], genes reported to be regulated by SREBP2 (*PCSK9*, *HMGCR*)[6,18,27], genes/proteins regulated by TCPOBOP in mice with 1% cholesterol diet (*Srebp2*, *Hmgcr*, *Ldlr*, *Insig1*, *Cyp7a1*, *Abcc4* and *Ugt2b34/UGT1A1 mRNA*)[8,26] and on INSIG1/Insig1 reported as a common PXR and Car target[11,28].

*SQLE* mRNA was found to be weakly but significantly upregulated by rifampicin (5 μM), and the induction is abrogated by MI-883 at 1 μM (Fig. 3f) and SPA70, but only SPA70 downregulated the basal expression of *SQLE* mRNA in HepaRG cells (Fig. 3f; Supplementary Fig. 10a). In the case of *SREBF1*, a CAR-dependent downregulation of mRNA expression was observed by MI-883 (Fig. 3f). MI-883 had no significant effect on *SREBF2* mRNA expression. However, SREBPs activities are mainly regulated at the post-transcriptional level in Golgi[27]. MI-883 significantly upregulated *INSIG1* mRNA, but likely both receptors are involved in the gene regulation (Fig. 3f).

We observed significant PXR-dependent upregulation of *LDLR* mRNA after treatment with MI-883 suggesting that PXR antagonism is involved in the gene regulation in HepaRG cells (Fig. 3g). Marginal effect of rifampicin on *PCSK9* mRNA (Supplementary Fig. 10b) suggests that PCSK9 gene is rather regulated by mature nuclear Srepb2 through the SRE promoter element than as a direct PXR or CAR target[11,27]. In the case of CYP7A1, the rate-limiting enzyme for bile acid production, PXR agonist rifampicin downregulated *CYP7A1* mRNA. Unexpectedly, we also observed significant downregulation of *CYP7A1* mRNA expression after treatment with MI-883 in the absence and presence of rifampicin (Fig. 3g). MI-883 induced *CYP7A1* mRNA in HepaRG KO CAR cells treated with MI-883. These results suggest that the PXR antagonism activity of MI-883 may upregulate *CYP7A1* mRNA expression in the absence of CAR, but this data needs further mechanistic investigation. In the cases of *ABCC4* and *UGT1A1* genes, they were upregulated by MI-883, indicating the dominant role of activated CAR over PXR inhibition in the gene's regulation (Supplementary Fig. 10).

In 2D PHH (OFA donor), MI-883 induced *CYP2B6* and *LDLR* mRNA but downregulated *CYP7A1* mRNA (Fig. 3h), other effects were minimal.

These results suggest unknown complexities in the regulation of these genes, mainly in the case of the human *CYP7A1* gene, which needs further extensive study in human hepatocyte models.

## MI-883 selectively decreases plasma cholesterol concentration

In the proof-of-concept experiments, we investigated the metabolic effects of MI-883 in humanized PXR-CAR-CYP3A4/3A7 mice fed either with a high-fat diet (HFD) to induce diet-induced metabolic syndrome with hypercholesterolemia or with a high-cholesterol diet (HCD) to induce diet-induced hypercholesterolemia (Fig. 4a, f).

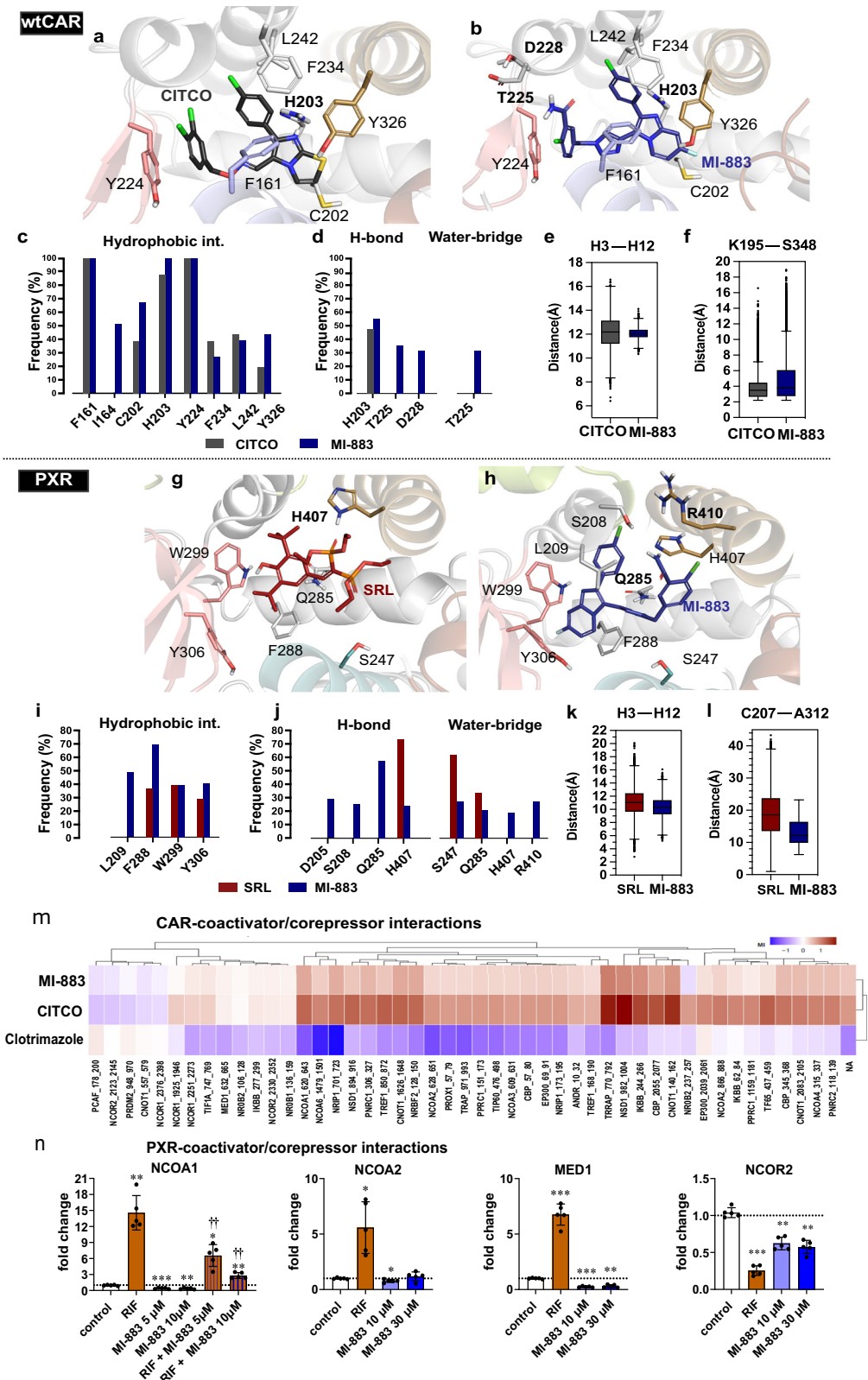

In the first model, MI-883 had no significant effect on body weight or food intake (Fig. 4b, c; Supplementary Fig. 12c). MI-883 produced an increase in liver weight (LW) and liver-to-body weight (LW/BW) after 1-month application of MI-883 (Fig. 4b), but not after five doses (*p.o.* 5 mg/kg, five doses every other day) in mice with HCD (Fig. 4g). This is consistent with the Car-mediated centrilobular hypertrophic but not proliferative (connected with replicative DNA synthesis) effects of

human CAR ligands in humanized CAR mice[7]. Importantly, we did not observe affected hepatic histology, morphology, and steatosis with ballooning cells after treatment with MI-883 (Fig. 4d), or any signs of hepatotoxicity or nephrotoxicity based on biochemical parameters in both studies (Supplementary Fig. 11).

We observed a systematic decrease in total plasma cholesterol concentration and LDL cholesterol levels after treatment with MI-883

**Fig. 2 | Molecular dynamics (MD) simulations analysis of wtCAR-LBD complexed with CITCO and MI-883, and PXR-LBD complexed with SR12813 and MI-883. a, b** Proposed binding mode of CITCO and MI-883 in the LBD of wtCAR represented by the most populated cluster conformation from the MD simulations, performed using hierarchical clustering of the backbone atoms from the aligned trajectory. Residues performing H-bond are labeled in bold. **c** Protein–ligand hydrophobic interactions include: π–π contacts and hydrophobic interactions. **d** Protein–ligand hydrogen bond (H-bond) and water-mediated interactions. Each simulation replica runs for 1 microsecond (μs) (5 independent replicas were conducted). **e** Distance between Helix3 (H3, center of mass of residues 157-178) and Helix12 (H12, center of mass of residues 341-348) in CAR-CITCO and CAR-MI 883. Boxplots span from 25% to 75% quartiles (IQR: 25–75%); the black line is the median. Minimum and maximum are the smallest and the largest data points, respectively. Whiskers are the data within 1.5*IQ. Outliers are indicated with diamonds. Distances were monitored each ns, i.e., there are: ∼5000 individual datapoints for each ligand. Full data is available in Zenodo repository. **f** Distance between K195 (side chain) and S348 (oxygen atoms). **g, h** A representative snapshot of PXR-LBP with SR12813 and MI-883. **i** Protein–ligand hydrophobic interactions include: π–π contacts and hydrophobic interactions. **j** Protein–ligand H-bond and water-mediated interaction. **k** Box plot represents the distribution of the distance between the center of mass of Helix3 (H3, residues 240–260) and Helix 12 (H12, residues 423–430). **l** Cα–Cα distance between C207 (of H2′) and A312 (of β4-H6 loop). **m** Heatmap of MARCoNI array data indicating interactions of human CAR with coactivators and corepressors in the presence of CITCO, MI-883, and clotrimazole, an inverse agonist (all at 100 μM). Modulation index (MI) is the $\log_{10}$-transformed relative binding value, which is calculated by the compound's binding value, relative to the vehicle control. red color -stimulated interaction, blue color -release of interaction. **n** Mammalian two-hybrid assays for PXR/coactivator NCOA1, NCOA2, and MED1 recruitment and PXR/corepressor NCOR2 release. Transfected cells were treated with vehicle or 10 μM rifampicin (RIF) and/or the indicated concentrations of MI-883 for 24 h ($n = 5$). *$P$ value < 0.05; ** ††$P$ value < 0.01; ***$P$ value < 0.001; statistically significant effects $vs.$ control (*) or $vs.$ rifampicin-mediated effect (†) (ANOVA). Data represent means ± S.D.

(Fig. 4e, h). We also noticed a decrease in high-density lipoprotein (HDL) cholesterol concentrations in males in the experiments (Fig. 4e, h). This was not unexpected considering that mice have different lipoprotein metabolism due to the absence of cholesteryl ester transfer protein (CETP). Thus, mice distribute cholesterol dominantly into the HDL lipoprotein pool and to a lesser extent into LDL lipoproteins[29]. Unfortunately, humanized CAR/PXR mice with humanized lipoprotein spectrum (such as APOE*3-Leiden mice) are not available now. We did not see any significant effects on triglycerides and glucose concentrations in males (Fig. 4e, h; Supplementary Fig. 12b) in both studies. In females, we observed some increase in triglyceride levels in the second study, which was at the level of statistical significance. MI-883 treatment did not impact other biochemical parameters in the studies (Supplementary Fig. 11).

In the next experiments, we used a small group of the PXB-mouse humanized liver mice ($n = 5$) carrying human hepatocytes with an estimated replacement index of 70% or more in the liver[30]. The model thus expresses human CAR and PXR receptors in the liver and the human hepatocytes form VLDL, LDL, and HDL lipoproteins with cholesterol distribution in LDL, VLDL, and HDL, and triglyceride distribution in VLDL and LDL lipoproteins as in humans[31] (Supplementary Fig. 15). We did not observe any increase of human albumin or human ALT (h-ALT1) after one-week repeated application of MI-883 (2.5 mg/kg per day). The effect of MI-883 on the total cholesterol concentration in the study did not reach statistical significance ($n = 5$) even though there may be a tendency for a decrease in LDL and an increase in HDL cholesterol plasma concentrations (Supplementary Fig. 15).

We also examined the influence of MI-883 on glucose homeostasis, glycemia, insulin levels, and insulin resistance in humanized PXR-CAR-CYP3A4/3A7 mice. We found that MI-883 had no effect on plasma glucose levels in both studies. Specifically, MI-883 did not significantly decrease postprandial glucose concentrations in the intraperitoneal glucose tolerance test (IPGTT) or in the intraperitoneal insulin tolerance test (IPITT). Additionally, we observed no effect of MI-883 on insulinemia (Fig. 4h, Supplementary Fig. 12d–f).

### Analyses of cholesterol, bile acids, and lipidome

Employing lipidomic and metabolomic tools, we found that free cholesterol (calculated per liver protein or LW) significantly decreased in the liver of mice treated with MI-883 in the HFD study, whereas lathosterol, total cholesterol, and cholesteryl esters (CE) did not show significant changes (Fig. 5a). C4 (7α-hydroxy-4-cholestene-3-one), the precursor of cholic acid (CA) and chenodeoxycholic acid (CDCA) synthesis, and C4/total cholesterol did not increase (Fig. 5b). Using UHPSFC/MS lipidomic analysis, we found that cholesteryl esters (CE) are not significantly altered in the livers after MI-883 treatment (Fig. 5i, k).

Employing biochemical and metabolomic analyses, we found that the total bile acid concentration was not affected after treatment with MI-883 in the plasma and livers in the HFD study (Fig. 5c, d, for hepatic bile acid data, see Supplementary Fig. 16). However, observed a significant increase in the total BA content in the stool of MI-883-treated mice after 24 h, including a significant ($P < 0.05$) increase in βMCA and CDCA (Fig. 5e, f). Deoxycholic acid (DCA), the most important secondary bile acid in mice produced by bacterial 7α-dehydroxylation, along with deconjugated α/βMCA, were the most abundant BAs in feces (Fig. 5f). We did not observe any diarrhea in the MI-883-treated animals.

We observed no significant impact of MI-883 on the TG accumulation in male livers using an enzymatic assay (Fig. 5g). However, closer analysis employing UHPSFC/MS lipidomics approach revealed that there is an intra-class effect in TG class, characterized by a significant increase of triglycerides containing long and highly unsaturated fatty acids (TG 60:12, TG 60:13) compared to vehicle-treated male mice (Fig. 5h, j). Additionally, we found a statistically significant decrease ($P < 0.05$) in the total and some individual lysophosphatidylcholine (LPC) levels (Fig. 5i, l).

We also performed metabolomic analysis of plasma samples from male hPXR/hCAR/hCYP3A4/3A7 mice treated with MI-883. Principal component analysis and a heatmap of metabolite level changes suggest no significant effect of MI-883 on plasma small-molecule metabolome (Supplementary Fig. 18, 19). Only three metabolites were significantly altered by MI-883 treatment: isoleucine (1.638-fold, $P = 0.0285$), 2-hydroxy-3-oxopropanoate (1.595-fold, $P = 0.0499$), and 5-aminopentanoic acid (or 5-aminovalerate) (1.375-fold, $P = 0.018$).

Overall, we determined that MI-883 decreases free cholesterol in the liver and increases fecal excretion of BAs, but it does not affect plasma or liver levels of triglycerides, bile acids, or the plasma metabolome.

### NGS RNA-Seq of liver samples

Next, we performed NGS RNA-Seq analyses of liver samples in the proof-of-concept study with MI-883 administered to humanized PXR-CAR-CYP3A4/3A7 animals fed with HFD (Fig. 4a). Unsurprisingly, we found that MI-883 significantly upregulated some genes and pathways involved in xenobiotic metabolism and the lipid metabolism of unsaturated fatty acids and eicosanoids (Fig. 6b–d). Conversely, the analysis showed that MI-883 significantly downregulated several genes involved in fatty acid metabolism (such as *Scd1*, *Fabp2*, and *Srebf1*) (Fig. 6c,d, Supplementary Fig. 13).

A detailed examination of the RNA-Seq data also showed that MI-883 marginally, but significantly, downregulated *Scarb1*, a gene encoding HDL receptor in the liver; *Angptl3*, an inhibitor of lipoprotein and endothelial lipases; and *Lipc* gene encoding the hepatic lipase (Fig. 6e). Additionally, some genes for apoproteins, such as *Apoc1*,

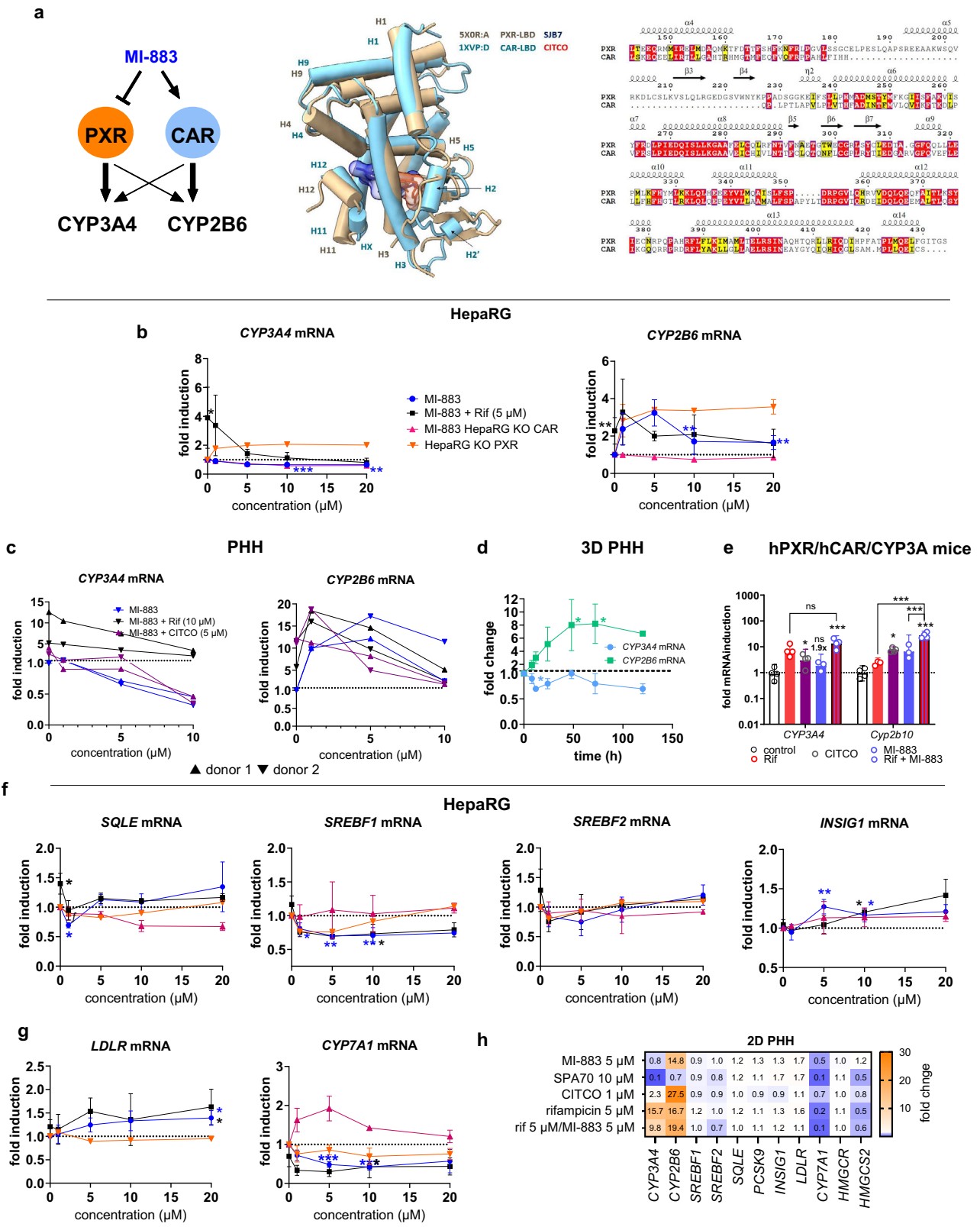

*Apoc4, Apoa5*, and *Apoa1*, were marginally but significantly down-regulated, likely due to Car activation[32] (Fig. 6d).

Employing RT-qPCR analysis, we confirmed the NGS RNA-Seq data for major target CAR/PXR genes including *Abcc4* encoding the Mrp4 efflux transporter, and genes encoding conjugation enzymes such as *Ugt1a1, Ugt2b34*, and *Sult2a1* (Fig. 6f). Importantly, Ugt1a1, Ugt2c34, and Sult2a1 are involved in conjugation of bile acids with sulfate and

glucuronide in mice being a minority elimination pathway[33–35]. Interestingly, we found significant upregulation of cholesterol 7α-hydroxylase (Cyp7a1) and sterol 12α-hydroxylase (Cyp8b1) proteins, but a downregulation of *Cyp7a1* mRNA (Fig. 6g, h). These enzymes are the rate-limiting steps in the classical pathways of bile acid synthesis from cholesterol[33]. *Hmgcr* gene, encoding for HMG-CoA reductase, the rate-controlling enzyme of the mevalonate pathway, was slightly

**Fig. 3 | Demonstration of CAR agonistic/PXR antagonistic activities of MI-883 in HepaRG cells, 2D and 3D primary human hepatocytes, and humanized PXR-CAR-CYP3A4/3A7 mice. a** CAR and PXR superimposition demonstrate the similarity of the related receptors of the NR1I group and their ligand binding domains (LBDs). Induction of *CYP3A4* mRNA and *CYP2B6* mRNA **b** expression in HepaRG, HepaRG KO PXR, HepaRG KO CAR cells or **c** 2D primary human hepatocytes (PHH, Biopredic, two donors) after treatment with MI-883 (range from 1 to 10 or 20 μM) for 24 h. Each analysis in HepaRG cells was done using independent triplicates; data in PHH represent a mean of technical triplicates from each donor. **d** Induction of *CYP3A4* and *CYP2B6* mRNA in 3D human hepatocyte spheroids (3D PHH) from three donors after treatment with MI-883 (5 μM) over 120 h in culture. Data are means ±S.D. from 3D hepatocyte cultures from tree donors. **e** Hepatic induction of *CYP3A4* and *Cyp2b10* mRNA in humanized PXR-CAR-CYP3A4/3A7 mice after *i.p.* treatment

of randomly assigned animal groups ($n$ = 3-4) with the vehicle, rifampicin (Rif, 3×5 mg/kg, every 24 h for 3 days), MI-883 (10 mg/kg every 24 h for 3 days) or their combination. CITCO was applied at 10 mg/kg in two doses after 24 h. Data in bar charts represent means±S.D. **f, g** mRNA expression of key genes involved in cholesterol synthesis (*SQLE*), cholesterol metabolism regulation (*SREBF1, SREBF2, INSIG1, LDLR*), and in bile acid synthesis (*CYP7A1*) in HepaRG, HepaRG KO CAR or HepaRG KO PXR cells, and **h** 2D PHHs (OFA donor) after treatment with CITCO (1 μM), SPA70 (10 μM) and MI-883 (5 μM) in the presence or absence of rifampicin 5 μM (rif) for 48 h. All experiments with HepaRG cell line have been done at least in three independent triplicates ($n$ = 3). Data represent means ± S.D. *$P$ value < 0.05; ***$P$ value < 0.001; statistically significant effects *vs.* control (*) or *vs.* rifampicin-mediated (*f*) effect in HepaRG cells or animals (ANOVA with multiple comparisons). *ns* - nonsignificant.

upregulated after treatment with MI-883 (1.31-fold upregulation) (Fig. 6e). Cyp2c70, a mouse enzyme that converts CDCA to αMCA and UDCA to βMCA, has been slightly but significantly downregulated in the study (−0.58 log2FC, 0.008 FDR_DESeq) (Fig. 6d). Transporters eliminating bile acids or cholesterol from hepatocytes into bile (*Abcb1, Bsep*, and *Abcg5/8*) have not been significantly affected (Fig. 6d).

Next, we focused on genes involved in the regulation of cholesterol synthesis and homeostasis. *Srebf1* gene, which encodes a key transcription regulator in lipid and cholesterol synthesis, was significantly downregulated (Fig. 6d, e, i). This effect is not the consequence of fasting, as mice treated with MI-883 have a similar food intake as control mice (Fig. 4c)[36]. Sterol regulatory element binding protein-1 (SREBP-1), a transcription factor with a basic helix–loop–helix leucine zipper, has two isoforms, Srebp-1a and Srebp-1c, both derived from the same gene for regulation of lipogenesis genes. This activity is critically dependent on its nuclear protein fragment, which is released from the endoplasmic reticulum via complex feedback machinery triggered by membrane cholesterol depletion[6,27]. However, we did not see any significant decrease in 120 kDa protein or its regulatory mature nuclear fragments (mSrebp1) with 68 kDa in the livers of MI-883-treated animals (Fig. 6i). In the case of Srebp2, the most important regulator of cholesterol synthesis and homeostasis, we observed that mature nuclear mSrepb2 protein may decrease in the livers after treatment with MI-883, although the effect is not significant (Fig. 6i). In addition, we observed a potent and significant ($P$ = 0.02) upregulation of Insig1 protein, but not *Insig1* mRNA, which is the critical negative regulator of Srebps in cholesterol synthesis regulation (Fig. 6j). Thus, we hypothesize that MI-883 suppresses Srebp2 activity through Insig1 elevation, however, further direct mechanistic evidence showing lower Srebp2 transcriptional activity is needed. (Fig. 5i,l).

Upon reanalyzing hepatocyte gene expression critical for liver steatosis and hepatic lipogenesis using RT-qPCR ($n$ = 11), we confirmed the NGS RNA Seq data showing that stearoyl-CoA desaturase (*Scd1*), which encodes the rate-limiting enzyme in the formation of mono-unsaturated fatty acids, is significantly downregulated ($P$ = 0.0036) following treatment with MI-883. Additionally, we found that fatty acid synthase (*Fasn*), primarily involved in palmitate synthesis, and *Plin2* mRNA (which encodes perilipin 2, crucial for lipid droplet formation) were also significantly downregulated ($P$ = 0.007 and $P$ = 0.018, respectively) (Supplementary Fig. 13). We also investigated whether MI-883 regulates key ileal transporters of BAs reabsorption. We found that the apical sodium-dependent bile acid transporter Asbt/Ibat (Slc10a2) of conjugated bile acids and organic solute transporter, alpha subunit (Slc51a) are significantly downregulated at the mRNA level after treatment with MI-883 and that Asbt protein is also significantly decreased in the mouse ileum (Fig. 6k). These results correspond with the effect of TCPOBOP on these BA transporters involved in BAs enterohepatic circulation[9]. We did not observe any regulation of NPC1-like intracellular cholesterol transporter 1 (Npc1l1), which plays a key role in intestinal cholesterol absorption, nor of the Abcg5/8 transporter, which is involved in transintestinal cholesterol excretion

(TICE), in the ileum (Supplementary Fig. 14). Additionally, *Shp* and *Fgf15* mRNA levels were not affected by MI-883 treatment in the ileum (Supplementary Fig. 14).

## MI-883 does not stimulate human hepatocyte hypertrophy and hyperplasia

Rodent Car activation is known to be associated with liver hypertrophy and hyperplasia due to hepatocyte proliferation and replicative DNA synthesis through CAR interaction with YAP signaling. However, this phenomenon is diminished in humanized CAR mice or absent in humanized liver mice with engrafted human hepatocytes[7,37,38]. In PXR-CAR-CYP3A4/3A7 mice, we observed statistically significant upregulation of murine *Gadd45b* mRNA and Meg3 lncRNA, both of which are antiapoptotic genes (RNA genes) (Fig. 7a).

In contrast, we found no significant effects on critical proliferative biomarkers MDM2, PCNA, and FOXM1 in both 3D human hepatocyte spheroid (Fig. 7b) and in PXB humanized liver mice with most human hepatocytes in the livers (Fig. 7d). Consistent with these results, we did not observe significant liver enlargement (LW/BW ration, ratio of liver weight to body weight) in PXB humanized liver mice (Supplementary Fig. 15). We also did not see any decrease in spheroid size or ATP production after MI-883 treatment in 3D primary human hepatocytes (PHHs) (Fig. 7c). Notably, we observed upregulation of *CYP2B6* mRNA, but not statistically significant induction of *CYP3A4* mRNA in PXB humanized mice (Supplementary Fig. 17).

## Discussion

The nuclear receptors PXR and CAR act as chemosensors and induce the transcription of a battery of target genes related to drug metabolism and clearance. However, recent studies indicate the important roles of CAR and PXR in the regulation of intermediary metabolism.

Activation of Car in mice has been shown to mitigate hepatic steatosis, increase glucose tolerance and insulin sensitivity, and reduce obesity under metabolic and nutritional stress[39,40]. Activation of PXR has been demonstrated to increase lipogenesis, reduce lipid oxidation, and promote a fatty liver phenotype in various animal and cellular models, as well as in clinical trials[14,20,41–43].

Distinct effects of CAR[8–10] and PXR[6,11] activation in the regulation of cholesterol homeostasis are challenging to discern, as simultaneous CAR activation and PXR antagonism may exhibit additive effects in the treatment of pathological hypercholesterolemia or other metabolic diseases. Moreover, a recent report uncovered inhibitory interaction between PXR and CAR[44]. Despite these insights, there has been no effort so far to design a combined CAR agonist/PXR antagonist.

Structurally, PXR and CAR are sibling receptors of the group NR1I, sharing a common ancestor. Consistently, their LBDs and cavities share many spatial and structural similarities (Fig. 3a). Therefore, it is challenging to identify a specific human CAR agonist without off-target PXR activation or to develop a high specific PXR antagonist[19,45,46]. Hydrophobic and promiscuous ligand-binding cavities of the receptors are the reason why CAR ligands usually coactivate PXR. Moreover,

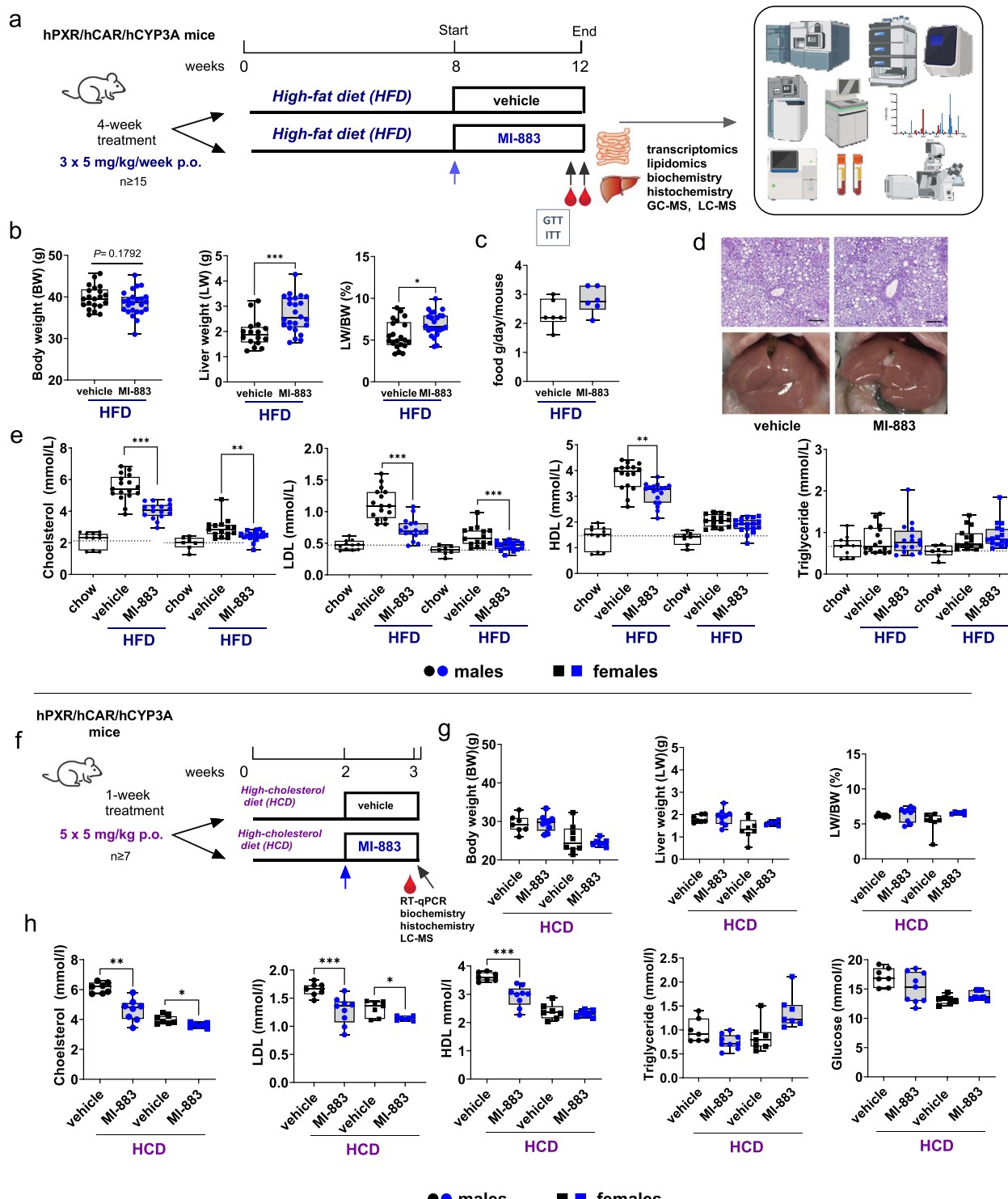

**Fig. 4 | Metabolic effects of MI-883 in the model of diet-induced hypercholesterolemia with high-fat diet (HFD) or high-cholesterol diet (HCD). a** Both male and female humanized PXR-CAR-CYP3A4/3A7 mice were fed for 8 weeks with HFD and treated with MI-883 (5 mg/kg 3× per week) via *p.o.* gavage for one month ($n \geq 15$). In parallel, control group were treated with vehicle via *p.o.* gavage. **b** Body weight (BW), liver weight (LW), LW/BW ratio, and **c** food intake were monitored. **d** Representative microscopic (H&E staining, bar 100 μm) and macroscopic pictures of the livers in the vehicle- or MI-883-treated mice fed with HFDs. **e** Plasma total cholesterol, LDL cholesterol, HDL cholesterol, and triglyceride levels were assessed at the end of the studies after 3 h of fasting. **f** Both male and female humanized PXR-CAR-CYP3A4/3A7 mice were fed for two weeks with a high-cholesterol diet HCD ($n \geq 7$). MI-883 (5 mg/kg 3× per week) was applied via *p.o.*

gavage for 10 days. (5 mg/kg every other day). **g** Body weight (BW), liver weight (LW), LW/BW ratio were monitored. **h** Plasma total cholesterol, LDL cholesterol, HDL cholesterol, triglyceride, and glucose levels were assessed at the end of the studies after 3 h of fasting. In parallel, control groups were treated with vehicle via *p.o.* gavage. Animals were randomly and independently assigned to control or treatment groups before interventions. *$P$ value < 0.05; **$P$ value < 0.01; ***$P$ value < 0.001, the significant effect of MI-883 to vehicle-treated control mice (Mann–Whitney U test, two-sided). Data are presented in box-and-whisker representing the 25th to 75th percentiles (box) and median (line). Whiskers represent minimum to maximum values. The dotted lines represent average plasma concentrations in humanized PXR-CAR-CYP3A4/3A7 mice on a chow diet. The figure in panel **a** was created using the BioRender tool.

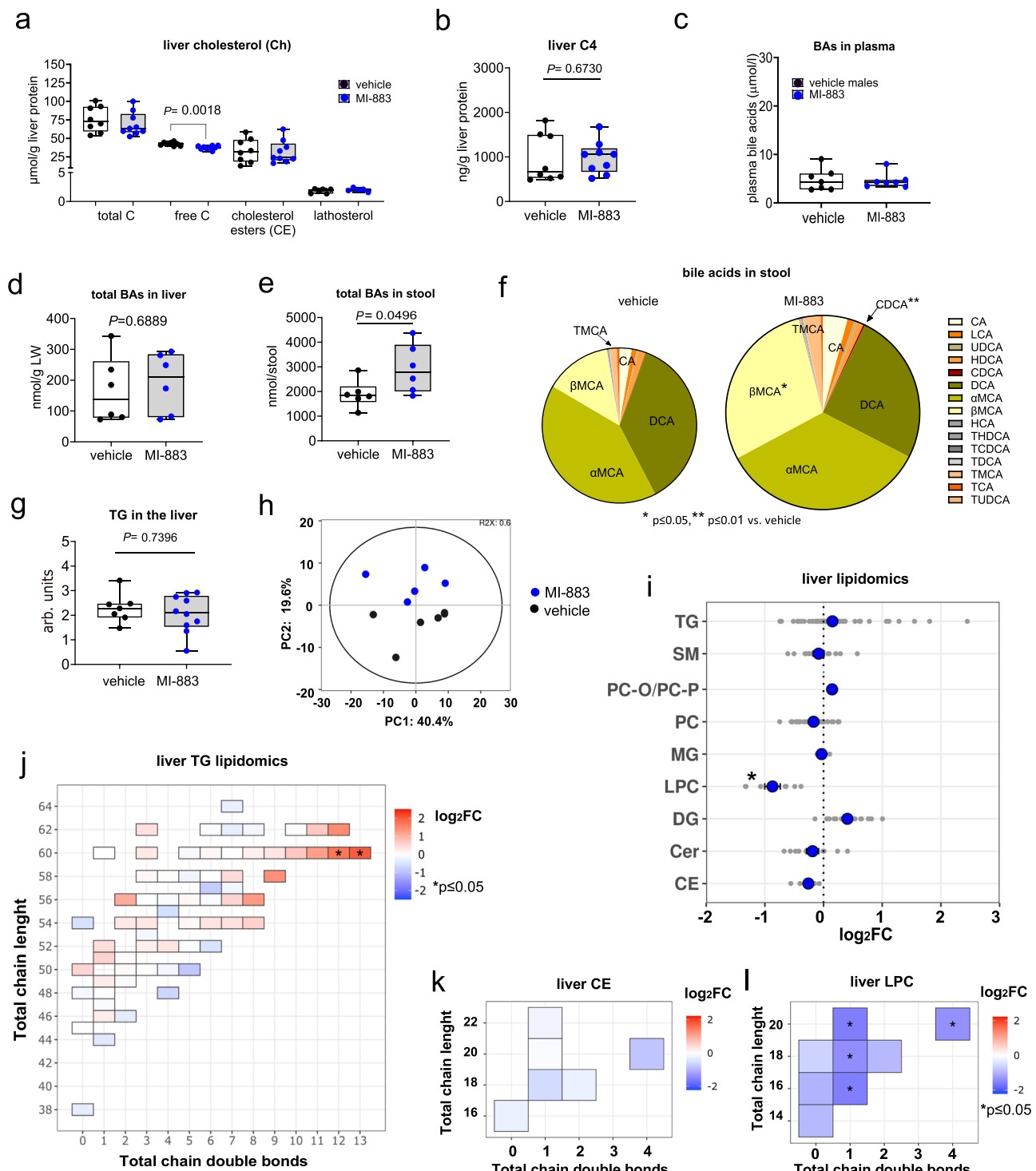

ligands for these nuclear receptors generally exhibit high lipophilicity and low metabolic stability, leading to unfavorable ADME properties and low potency at the receptors, typically observed at micromolar concentrations.

In this study, we discovered the CAR agonist/PXR antagonist exhibiting exceptional potency toward both receptors (Fig. 1). Notably, MI-883 demonstrates favorable physicochemical properties, metabolic stability, and pharmacokinetic profile characterized by optimal absorption and predominant distribution to the liver. Importantly, our findings reveal MI-883's high safety profile, as evidenced by its non-toxicity in 3D spheroids of primary human hepatocytes and in 28-day repeated oral dose toxicity studies in rats (Fig. 7c, Supplementary Chapter 18).

In cellular hepatic models, MI-883 showcases intriguing complementary activities in regulating key genes associated with cholesterol synthesis and homeostasis. Specifically, our results illustrate MI-883's ability to downregulate *SREBF1* mRNA in a CAR-dependent manner (Fig. 3f). Moreover, we observe that MI-883-mediated *LDLR* mRNA upregulation is contingent upon PXR antagonism, whereas INSIG1 induction is dependent on common modulation of CAR and PXR (Fig. 3f). Furthermore, our data indicate significant modulation of *CYP3A4* and *CYP2B6* gene expression by PXR inhibition, with PXR playing a dominant role in CYP3A4 regulation (Fig. 3b, c). Our investigations in 3D human hepatocyte spheroids (Fig. 3d) and PXB human liver mice (Supplementary Fig. 17) suggest that MI-833 is not a CYP3A4

**Fig. 5 | Lipidomic and bile acid analyses of cholesterol, bile acids, and lipids in plasma, stool, and livers of humanized PXR-CAR-CYP3A4/3A7 mice treated with MI-883 in the study with the high-fat diet. a** Total liver cholesterol (Ch) content, free cholesterol, cholesteryl esters (CE), cholesterol precursor lathosterol, and **b** bile acid precursor C4 contents were analyzed using LC-MS in the male liver samples ($n \geq 8$). **c** Plasma bile acid concentration in males ($n = 7$), **d** total liver bile acid (BA) content, and **e** total BA content in stool per 24 h of males hPXR/hCAR/hCYP3A male mice treated with MI-883 (both $n = 6$). **f** Bile acid analysis of individual acids in stool ($n = 6$). **g** Enzymatic analysis of triglyceride (TG) content in the livers ($n \geq 7$). TG and glycerol levels in samples were measured using enzymatic assays. Data are presented as the difference between TG and glycerol content after spectrophotometric detection in arbitrary units (arb. units). Lipidomic analysis in the livers of MI-883-treated animals. Lipidomics - **h** Principal component analysis (PCA) analysis, **i** triglycerides (TG), sphingomyelins (SM), ether/plasmalogen phosphatidylcholines (PC-O/PC-P), phosphatidylcholines (PC), monoglycerides (MG), lysophosphatidylcholines (LPC), diglycerides (DG), ceramides (Cer), and CE contents in the male liver homogenates. Data are presented as log2 fold changes ($\log_2FC$) of lipid compounds between the MI-883-treated and control groups, showing individual values (grey) and their means (blue) for each lipid class. **j** Analysis of fatty acid saturation in TG lipid class (total chain double bonds) in relation to the total chain length. **k** Fatty acid saturation (total chain double bonds) in relation to the total chain length in CE and **l** LPC lipid classes. Animals were randomly and independently assigned to control or treatment groups before interventions (five animals per groups, $n = 5$). Box-and-whisker plots represent the 25th to 75th percentiles (box) with median (line). Whiskers represent minimum to maximum values. *$P$ value < 0.05; statistically significant increase or decrease in lipid or individual bile acid content to control (vehicle-treated) animals (Mann–Whitney U test, two-sided).

inducer in humanized models, thus alleviating concerns regarding potential drug-drug interactions via this key metabolizing enzyme. Additionally, the regulation of classical CAR-target genes such as *UGT1A1* and *ABCC4*, and murine genes including *Sult2a1*, *Ugt1a1*, *Ugt2b34*, and *Abcc4*, remains unaffected by MI-883's PXR antagonism, underscoring the predominant role of CAR in their regulation[47] (Fig. 6f, Supplementary Figs. 10, 17).

Our animal studies with MI-883 recapitulated many features of TCPOBOP activity on plasma, liver, and overall body cholesterol content, as well as cholesterol and BA homeostasis reported in wild-type, *ApoE*[-/-] and *Ldlr*[-/-] mice[9,10,32]. Specifically, we observed: (i) increase expression of important conjugation enzymes and efflux transporters of BAs in the liver, (ii) increased expression of Cyp7a1 and Cyp8b1 enzymes in mice which promote the conversion of cholesterol into BAs, (iii) downregulation of Asbt and Ostα transporters in the ileum, which control the reabsorption of BAs, (iv) a significant increase in BAs content in the fecal matter, (v) decreased free cholesterol levels in the liver, (vi) unaffected nuclear Srebp1 and Srebp2 transcription factors, indicating a disrupted feedback response to reduced cholesterol levels in the liver and plasma, (vii) significant upregulation of Insig1, a negative regulator of Srebps by MI-883, (viii) minor but systematic downregulation of both proatherogenic and antiatherogenic apolipoproteins[48] in the liver, and (ix) downregulation of *Angplt3* mRNA after treatment with MI-883. Notably, *Angptl3* gene expression is associated with a substantial reduction in plasma cholesterol and triglyceride levels in mice or individuals with loss-of-function (LOF) variants[49].

Moreover, plasma triglycerides, glucose, urea, uric acid, iron, ions, and other biomarkers, as well as plasma metabolites examined using metabolomic analysis, remained largely unchanged after MI-883 treatment, indicating a specific effect on cholesterol and BA homeostasis (Fig. 4, Supplementary Figs. 11, 12, 18). Importantly, none of the tested hepatic biomarkers for liver injury exhibited significant alterations following one-month treatment with MI-883 (Supplementary Fig. 11), and we did not observe any significant upregulation of biomarkers for hepatocyte proliferation in human cellular models or in the humanized livers of PXB mice (Fig. 7).

Recent studies have shown that TCPOBOP treatment significantly reduces two taxa in Bifidobacterium spp., which exhibit bile salt hydrolase (*bsh*) activity. Although the link between decreased *bsh* and the unconjugated BA pool in the colon has not confirmed[50], this finding warrants further investigation into the potential effect of MI-883 on gut microbiome as an important aspect of cholesterol/BA homeostasis.

We can conclude that this study underscores the viability of combined modulation of PXR and CAR receptors, enriching our understanding of their co-regulation of target genes involved in metabolism. Based on results from PXR-CAR-CYP3A4/3A7 mouse and human hepatocyte models, we show MI-883 as a prominent modulator of cholesterol homeostasis. Furthermore, it opens new avenues for interventions in metabolic diseases utilizing dual-acting ligands with favorable ADME properties.

## Methods

### TR-FRET CAR coactivator binding assay
The LanthaScreen TR-FRET CAR Coactivator Binding Assay Kit, goat (ThermoFisher Scientific, Cat. No PV4836) with GST-tagged human CAR ligand-binding domain (LBD) and a fluorescein-labeled PGC1α coactivator peptide was used as we have reported before[19].

The LanthaScreen TR-FRET PXR Competitive Binding Assay (ThermoFisher Scientific, USA) was performed to evaluate interaction with PXR LBD as we described[51]. The assay measures the ability of an evaluated compound to replace the fluorescent PXR ligand from the receptor. The fluorescence was measured using a Synergy 2 Multi-Mode Microplate Reader (BioTek Winooski, VT, USA) ($n = 3$).

Half maximal effective concentration to activate ($EC_{50}$) or inhibit ($IC_{50}$) CAR/PXR LBD in the assays was calculated from at least eight points (range of 10 pM to 30 μM) using the GraphPad Prism software.

### Cell lines
Human hepatocellular carcinoma HepG2 (RRID:CVCL_0027) and monkey fibroblast-like COS-1 (RRID:CVCL_0223) cell lines were cultured as described previously[19]. LS174T human colon adenocarcinoma cells (RRID:CVCL_1384) were used in induction experiments with the subsequent *CYP3A4* mRNA RT-qPCR analysis as they express functional endogenous PXR.

HepaRG cells from Biopredic (Rennes, France) were cultivated and differentiated on the 12-well plates. For each experiment, the HepaRG cells were seeded at the density of 26,600 cells/cm² and cultivated in William's medium supplemented with 5 μg/mL insulin, 50 μM hydrocortisone, 10% Hyclone fetal serum (GE Healthcare Life Sciences, Pittsburgh, USA). 14 days after seeding, the HepaRG cells (RRID:CVCL_9720) were differentiated into hepatocyte-like cells using 1.5% DMSO in culture media for another 14 days[52]. Several batches of HepaRG KO CAR (CAR Knockout HepaRG™ Cells, MTOX1012-1VL, RRID:CVCL_B6AA) and PXR Knockout HepaRG™ Cells, MTOX1011-1VL, RRID:CVCL_B6AS) cells have been obtained from Sigma-Aldrich/Merck (Darmstadt, Germany) and cultivated according to the provider's protocols. Induction experiments with HepaRG cells (24 or 48 h treatments) have been performed in duplicates in at least 3 independent experiments. Experiments with HepaRG KO PXR and HepaRG KO CAR cells have been performed in duplicates at least in two independent experiments. In Dr. Burk's laboratory, HepG2 cells (HB-8065, lot number 58341723, ATCC, Manassas, VA) were cultivated in minimal essential medium, supplemented with 10% FBS, 2 mM L-glutamine, 100 μ/ml penicillin and 100 μg/ml streptomycin. In transfection experiments in HepG2 cells, regular FBS was replaced by dextran-coated charcoal-treated FBS. HepG2 cells were obtained at passage 74, propagated, and used in transfection experiments between passages 99 and 111. The cells were routinely checked for the presence of mycoplasma by PCR (VenorGeM Classic, Minerva Biolabs, Berlin, Germany).

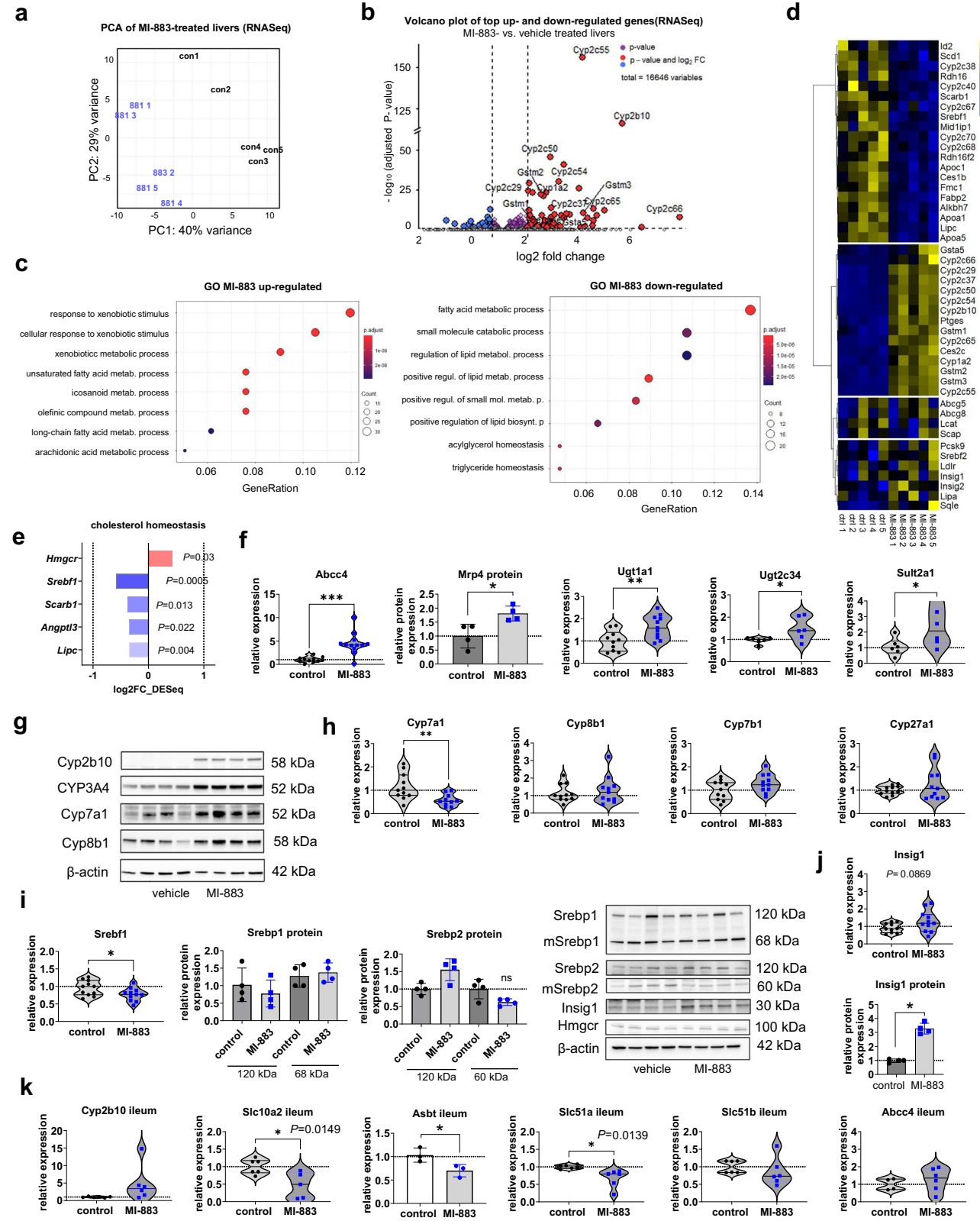

## CAR and PXR luciferase gene reporter assays

CAR3 luciferase reporter assay (with CYP2B6-luc) was performed in human hepatocellular carcinoma HepG2 cells as we described[19] For the CAR3 assay (Fig. 1d), a CYP2B6-luc reporter plasmid (originally entitled B-1.6k/PB/XREM) was kindly donated by Dr. Hongbing Wang (University of Maryland School of Pharmacy, Baltimore, MD, USA). All expression vectors are described in Supplementary, Chapter 4).

The CYP3A4 promoter luciferase reporter construct with a distal XREM (−7836/−7208) and a basal promoter sequence (prPXRE, −362/+53) from the CYP3A4 gene promoter region (CYP3A4-luc) was used for the PXR-dependent luciferase reporter gene assays (Fig. 1i, done at Charles University).

CYP3A4 enhancer/promoter-reporter gene plasmid pGL4-CYP3A4(-7830Δ7208-364) used at Dr. Margarete Fischer-Bosch-Institute of

**Fig. 6 | Next-generation sequencing transcriptomic, RT-qPCR, and Western blotting data of liver samples in humanized PXR-CAR-CYP3A4/3A7 male mice treated with MI-883 in the study with the high-fat diet.** NGS RNA-Seq transcriptomic data. **a** Principal component analysis (PCA) analysis, **b** volcano plot and **c** GO pathway analyses of top eight up-regulated and down-regulated pathways, and **d** heatmap of genes significantly regulated including the key genes of cholesterol homeostasis. **e** Detailed RNA-seq data of selected genes involved in cholesterol homeostasis significantly impacted by MI-883. **f** Expression of Mrp4 transporter, conjugation enzymes, and **g, h** enzymes involved in bile acid synthesis analyzed using RT-qPCR or Western blotting in the livers of MI-883-treated and control mice. **i** Expression of *Srebf1*/Srepb1 and *Srepf2*/Srepb2 mRNA and proteins in the livers. Western blotting of Srebp1 and Srebp2 proteins in the livers using the antibody detecting both precursor and mature nuclear forms (mSrebp−1c and mSrebp2). **j** Expression of *Insig1* mRNA and protein employing RT-qPCR and

Western blotting in the livers of MI-883-treated and control mice. **k** RT-qPCR and Wester blotting expression analysis of CAR target genes and bile acid transporters in the ileum of humanized PXR-CAR-CYP3A4/3A7 male mice in the model of HFD diet-induced hypercholesterolemia. Animals were randomly and independently assigned to control or treatment groups before interventions. For RNA-seq transcriptomic analysis, five independent liver samples from five animals per group ($n = 5$) were used. For RT-qPCR, at least ten animals ($n \geq 10$) were analyzed in each group. Western blotting analyses were performed with at least four independent samples from four animals ($n = 4$). Violin plots represent the frequency distribution curve with the median (central line) and two quartile lines. Data in bar charts represent means ± S.D. *$P < 0.05$; **$P < 0.01$; ***$P < 0.001$; statistically significant expression when compared to control (vehicle-treated) animals (Mann–Whitney U test, two-sided).

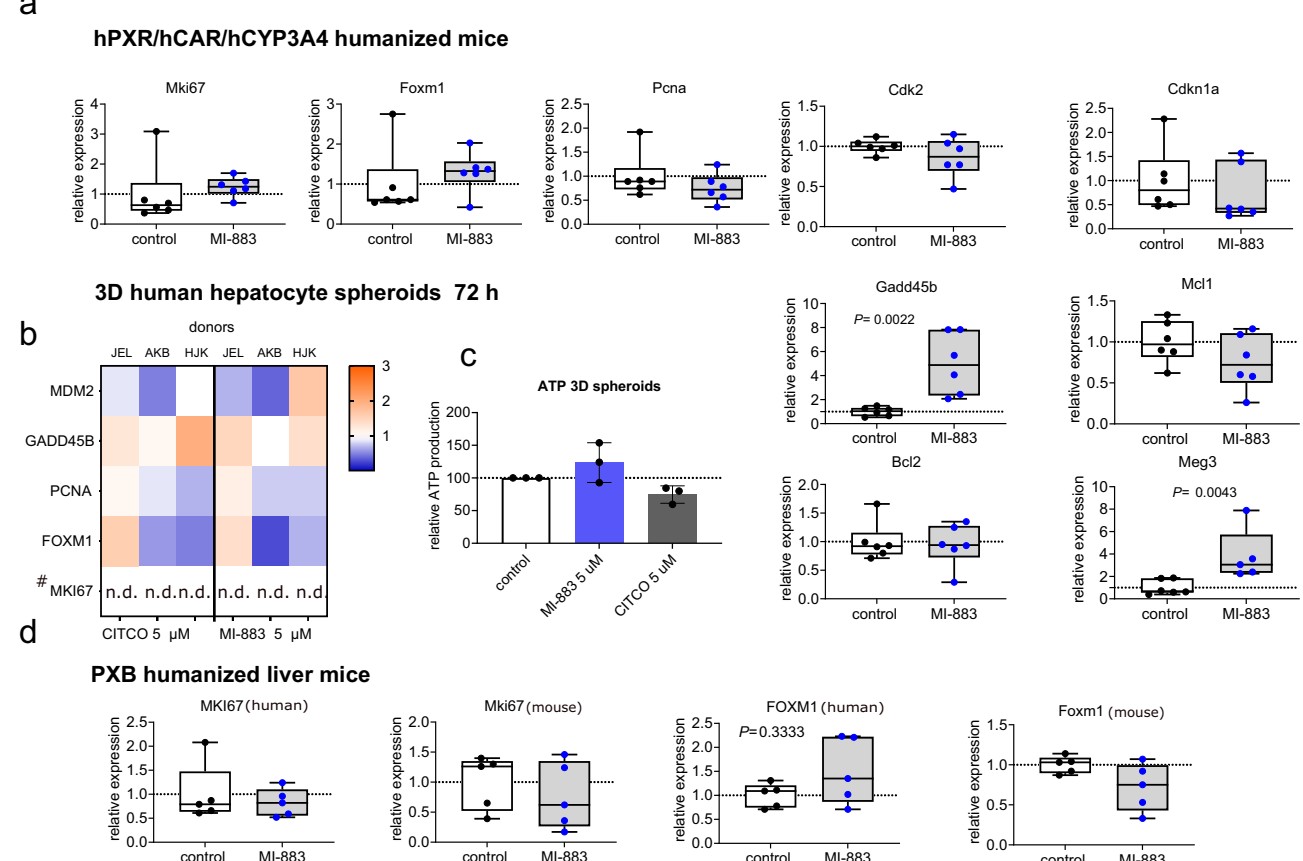

**Fig. 7 | MI-883 does not up-regulate biomarkers of human hepatocyte proliferation in humanized PXR-CAR-CYP3A4/3A7 male mice, 3D spheroids of primary human hepatocytes, and PXB humanized liver mice.** RT-qPCR analyses of mRNA expression for pro-proliferative, antiapoptotic, and proapoptotic biomarkers in **a** humanized PXR-CAR-CYP3A4/3A7 mice ($n = 6$), **b, c** 3D spheroids of human hepatocytes from three donors (JEL, AKB, HLK) after 72 h treatment, and **d** PXB humanized liver mice ($n = 5$) with 70% population of liver parenchyma with human hepatocytes treated for seven days with MI-883. **c** Effects of MI-883 and CITCO (at 5 µM) on ATP production in 3D PHH spheroids from three donors after

72 h treatment. **d** Humanized PXR-CAR-CYP3A4/3A7 mice and PXB humanized liver were treated with MI-883 for 1 month (3 × 5 mg/kg per week *p.o.*) or 2.5 mg/kg every day for 7 consecutive days *p.o.* ($n = 5$ in each group). RT-qPCR analyses have been performed in technical triplicates of frozen liver or hepatocyte samples. Expression has been related to vehicle-treated samples (set to 1). *P* value, Mann-Whitney U test, two-sided. # gene with very low mRNA expression; *n.d.* - expression not determined - C_t expression has been detected after 35 cycles of PCR. Box-and-whisker plots show the median (line) with 25th to 75th percentiles. Whiskers represent minimum to maximum values.

Clinical Pharmacology was described before[53]. The luciferase reporter plasmid was used together with the human wild-type PXR expression vector or with expression vectors with mutated PXR LBD (Fig. 1j).

Cells were treated for 24 h with MI-883 in a range of concentration from 0.1 nM to 30 µM ($n = 3$). The half maximal effective concentration ($EC_{50}$) to activate CAR or inhibit PXR ($IC_{50}$) in the assays was calculated from at least eight points of the dose-response curves using the GraphPad Prism software.

## CAR- and PXR-LBD assembly assays, and mammalian two-hybrid cofactor interaction assays with PXR

A human CAR LBD assembly assay (Fig. 1c) was performed according to protocols published by Carazo and Pavek with two hybrid expression constructs encoding helices 3−12 (pCAR-C/VP16) and helix 1 (pCAR-N/GAL4) of the human CAR LBD[54]. Transient transfection assays were carried out in the HepG2 cells using Lipofectamine 3000 transfection reagent (ThermoFisher Sciences). Cells were seeded into 48-well plates

(4×10⁴ cells per well) and transfected with a pGL5-luc luciferase reporter construct (150 ng/ well, Promega), expression constructs pCAR-C/VP16 and pCAR-N/GAL4 (100 ng/well) and the *Renilla reniformis* luciferase transfection control plasmid (pRL-TK, Promega) (30 ng/well) 24 h later. Cells were maintained in a phenol red-free medium (200 μL) and treated with MI-883 and CITCO for 24 h. Luminescence activity in the cell lysate was measured using Dual-Luciferase Reporter Assay, Promega, Madison, WI, USA. The half-maximal effective concentration (EC₅₀) to activate CAR in the assay was calculated using the GraphPad Prism software.

A human PXR LBD assembly assay (Fig. 1k) was done with expression plasmids encoding fusion protein of GAL4-DNA binding domain (DBD) and helix 1 part of the LBD of PXR (residues 132–188) and fusion protein of VP16 activation domain (AD) and helices 2-12 of the PXR-LBD (residues 189–434), respectively, which have all been described previously[55]. Metridia luciferase expression plasmid pMetLuc2control was obtained by Takara-Clontech (Mountain View, CA, USA). Transient batch transfection of HepG2 cells was done in 96-well plates using Jet-PEI transfection reagent (Polyplus, Illkirch, France), as described previously[56]. The following amounts of plasmid DNA per well were used: 0.24 μg pGL4-G5 and 0.03 μg each of expression plasmids encoding GAL4-DBD/PXR(132-188) and VP16-AD/PXR(189-434). All transfections included 0.01 μg pMetLuc2control per well for normalization of firefly luciferase reporter activities. After overnight incubation with the transfection mixtures, cells were treated with chemicals in a volume of 100 μl of cell culture medium. 24 h later, the medium supernatant was removed, and cells were lysed with 50 μl 1x passive lysis buffer (Promega, Madison, WI, USA) per well. Firefly luciferase activity was measured in 10 μl of cell lysates, and Metridia luciferase activity was analyzed in 10 μl of medium supernatant, both as described before[56]. Results were normalized by dividing firefly luciferase activity by *Metridia* luciferase activity measured from the same well.

The mammalian two-hybrid PXR-cofactor interaction assays (Fig. 2o) were performed with expression plasmids encoding fusion proteins of GAL4-DNA binding domain (DBD) and the receptor-interacting domains (RID) of nuclear receptor coactivator (NCOA1/SRC-1) 1 (residues 583–783), NCOA2 (residues 583–779), MED1 (residues 527–774), and nuclear receptor corepressor 2 (NCOR2) (residues 1109–1330)[55], which have all been described previously. Expression plasmid encoding a fusion protein of VP16 activation domain (AD) and PXR-LBD (residues 108–434) has been used before[55]. The GAL4-dependent pGL4-G5, harboring a pentamer of GAL4 binding sites in front of the E1b promoter was described before[56].

Transient batch transfection of HepG2 cells was done as described above. The following amounts of plasmid DNA per well were used: PXR-corepressor release assay, 0.23 μg pGL4-G5, 0.03 μg each of expression plasmids encoding GAL4-DBD/NCOR2-RID and VP16-AD/PXR(108-434) and 0.015 μg RXRα expression plasmid; PXR-coactivator recruitment assays, 0.24 μg pGL4-G5, 0.03 μg of expression plasmid encoding respective GAL4-DBD/coactivator-RID and 0.03 μg expression plasmid encoding VP16-AD/PXR(108-434). All transfections included 0.01 μg pMetLuc2control per well for normalization of firefly luciferase reporter activities. After overnight incubation with the transfection mixtures, cells were treated with chemicals in a volume of 100 μl of cell culture medium. 24 h later, medium supernatant was removed, and cells were lysed with 50 μl 1x passive lysis buffer (Promega, Madison, WI, USA) per well. Firefly luciferase activity was measured in 10 μl of cell lysates, and Metridia luciferase activity was analyzed in 10 μl of medium supernatant, both as described before[56]. Results were normalized by dividing firefly luciferase activity by *Metridia* luciferase activity measured from the same well. Measurements of samples were repeated several times (n ≥ 3).

Luciferase reporter gene assays in PXR expressing H-P cells and mammalian two-hybrid reporter assays have been performed as described before[55,57].

The luciferase reporter gene assays for FXR, LXRα, LXRβ, THα, GRα, PPARα, PPARδ/β, PPARγ, AhR, VD, ERα, ERβ and mouse Car are described in our recent paper[19] (see Supplementary Chapter 7).

**Translocation assay**

Nuclear translocation of pEGFP-hCAR+Ala chimera in COS-1 SV40 transformed African green monkey kidney cells was performed as we have described (n = 3)[19].

**Interactions between the CAR-LBD and coregulator peptides (MARCoNI array)**

The ligand-modulated coregulator interactions with the HIS-tagged CAR-LBD (ThermoFisher Sciences, Cat. No PV4836) and anti-HIS antibody labeled with Alexa488 (Qiagen) was assessed using a PamChip® microarray that contains 154 coregulator-derived binding peptides, including the LXXLL coactivator motif or the LXXXIXXXL corepressor motif from 66 different coregulators (PamGene International B.V, Supplementary, Chapte 9)[58,59]. The reaction was performed in a fully automated microarray processing and fluorescent imaging platform (PamStation12) at 20 °C for 80 cycles (two cycles per minute). After removal of the unbound CAR-LBD receptor by washing each array with 25 μl Tris-buffered Saline (TBS) buffer, fluorescent images of the PamChip microarrays were obtained using the PamStation12, and then were analyzed for quantification of CAR binding using BioNavigator software (PamGene International B.V.). For data and statistical analysis, BioNavigator was used.

Modulation index (MI) is the log₁₀-transformed relative binding value, which is calculated by the compound's binding value, relative to the vehicle control (DMSO) binding value. Positive interaction indicates that ligands increase CAR–coregulator motif interactions, and the cut-off is MI > 0; the binding value, interaction indicates that ligands decrease CAR-coregulator motifs interactions and the cut-off is MI < 0; the binding value, >50; the relative binding value <1.

**Primary human hepatocytes (PHH) and 3D spheroids of PHH**
**2D primary human hepatocyte culture.** Primary human hepatocytes (PHHs, donor OFA, BioIVT, Westbury, New York, USA) were seeded onto rat tail collagen type I-coated 24-well plate (Corning, New York, USA) (350,000 cells per well) in PHH medium as described[60] with 10% fetal bovine serum (FBS, HyClone), which was exchanged for FBS-free PHH medium 2 h later when the cells attached to the plate surface. The next day, 2D PHH cells were exposed to either MI-883 (5 μM), SPA70 (10 μM), CITCO (1 μM), rifampicin (5 μM) or rifampicin (5 μM) together with MI-883 (5 μM) diluted in FBS-free PHH medium. After 48 h, the cells were harvested to 1 ml of QIAzol lysis reagent (Qiagen, Hilden, Germany) and further analyzed by RT-qPCR (Fig. 3h).

In addition, the primary human hepatocytes (human hepatocytes in monolayer-long term cultures) from two donors were obtained from Biopredic (Rennes, France) (batch HEP220969, 78-year-old female, Caucasian; HEP220971, 46-year-old male, Caucasian, HEP220976, 73-year-old male). The cells were cultivated as 2D monolayers according to the manufacturer's protocol. The hepatocytes were treated with rifampicin (10 μM), CITCO (5 μM), MI-883 (1, 5, and 10 μM, respectively), or their combinations for 24 h in the use medium (Fig. 1g HEP220969 performed in triplicate samples; Fig. 3c HEP220971 and HEP220976).

**3D primary human hepatocyte spheroids.** Primary human hepatocytes (PHHs, female donors JEL, HJK, and AKB, BioIVT, Westbury, New York, USA) were seeded onto ultra-low attachment 96-well plates (Corning, New York, USA) (1500 cells per well) and cultivated according to a previously published protocol[60]. Briefly, PHHs were cultivated for 6 days to generate compact spheroids by spontaneous self-aggregation and then half of PHH medium was exchanged for FBS-free PHH medium. 3D PHHs were treated with MI-883 (5 μM)

commenced from day 8 within a time range of 8–120 h (time points were 8, 12, 24, 48, 72, and 120 h). Conditioned FBS-free PHH medium was changed every 2–3 days of the treatment. 30 spheroids per each condition were collected to 1 ml of QIAzol lysis reagent (Qiagen, Hilden, Germany) for downstream RT-qPCR analysis. Data are presented as the mean ± SD of fold change expression gained from 3D spheroids of three different donors. ATP assay has been performed as described[60].

## RT-qPCR and Western blotting
RT-qPCR was used to examine gene expression in 2D and 3D PHH, HepaRG cells, or in mouse liver or intestine (ileum) samples as we have described before[60–62]. TaqMan probes for murine and human genes are listed in Table S3 in the *Supplementary Information*. Antibodies used for Western blotting have been described[63] or are listed in Table S4 (Supplementary Chapter 13).

## Animal experiments
All animal studies were performed in accordance with the EU Directive 2010/63/EU and they were approved by the Czech Central Commission for Animal Welfare. All animal experiments were performed in compliance with, and all animals received care according to the guidelines set by the Animal Welfare Body of the Czech Centre of Phenogenomics. Humanized PXR-CAR-CYP3A4/3A7 mice (model 11585, Taconic, Rensselaer, NY, USA) were housed in a temperature-controlled environment (22 ± 2 °C) with 12-h light/dark cycle and free access to water. Unless otherwise stated, mice were fed a chow diet (Altromin, Cat. No. 1314 Forti, Lage, Germany) *ad libitum*.

For experiment involving a HFD in the model of diet-induced metabolic syndrome and type 2 diabetes with hypercholesterolemia, both male and female mice at the age of 10–15 weeks were pretreated for 8 weeks with HFD (E15186 EF R/M with 30% fat, ssniff ®, Soest, Germany) containing 30.1% crude fat, 20.8% protein and 171 mg/kg of cholesterol, *ad libitum* (Fig. 4a). Mice were then randomly assigned to MI-883 or control groups (17 male animals per group for the control or MI-883-treated group, 15 female animals per group for control and MI-883 treated groups, respectively). They were administered via oral gavage with either MI-883 treatment (5 mg/kg, 3x per week on Mo, We, Fri) or vehicle (5%DMSO/5%PEG300/90%corn oil) while still fed with HFD. Mice were sacrificed after 4 weeks of treatment.

In parallel, two groups of male animals (10 per group) and female animals (7 per group) were fed a chow diet for blood biochemistry analysis.

Intraperitoneal glucose tolerance test (IPGTT) and Intraperitoneal Insulin Tolerance Test (IPITT) assays were performed one week before the end of the HFD study. For IPGTT, mice fasted for 18 h were intraperitoneally injected with 20% D-glucose solution (2 g/kg BW). Blood was sampled before the injection and 15, 30, 60, and 120 min after. For the IPITT assay, mice fasted for 4 h were administered with 0.75 IU insulin/kg BW. Blood was sampled before the injection and 15, 30, 45, 60, and 90 min after.

For experiments with a high-cholesterol diet (HCD), mice (18–26 weeks) were pretreated *ad libitum* for 2 weeks with diet containing 15% fat and 12.5 g/kg of cholesterol (E15103, ssniff, Soest, Germany) (Fig. 4f). After this period, animals were randomly assigned to two groups (control vehicle-treated and MI-883-treated animals, with 9 males and 7 females per group). Animals were dosed orally with MI-883 (5 mg/kg every other day—Monday, Wednesday, Friday, and the following Monday and Wednesday, for a total of 5 doses) or vehicle alone (5% DMSO/5% PEG300/90% corn oil) via oral gavage three times per week. The animals remained on the HCD during the treatment period. Mice were sacrificed after 5 treatments.

In induction studies with humanized PXR-CAR-CYP3A4/3A7 mice, 3 to 4 male animals fed a chow diet were treated with 10 mg/kg MI-883, 5 mg/kg rifampicin or their combination three times via intraperitoneal

(*i.p.*) injection every 24 h. CITCO was applied at two doses (10 mg/kg) after 24 h and livers were sampled 24 h after the last dose. In the induction experiments, urine was sampled from the gallbladder. All animals were euthanized 24 h after the last administration through cervical dislocation under isoflurane anesthesia.

Mice were weighted and tissues (livers, blood, duodenum, jejunum, ileum, white fat) were removed, weighted and snap frozen in liquid nitrogen and stored at −80 °C until analysis. Blood was collected from the retro-orbital sinus of mice under isoflurane anesthesia into heparin-coated tubes (Kabe Laboratortechnik GmbH, Nümbrecht-Elsenroth, Germany) after 3 h fasting and analyzed on AU480 biochemistry analyzer (Beckman Coulter, Brea, USA). Sample preparation for bile acid and cholesterol analyses from tissues or feces is described below. One of the liver lobes was fixed in formalin, embedded in paraffin blocks, sectioned, and subjected to H&E staining. Blood was analyzed as described in Supplementary, Chapter 10.

## Pharmacokinetic study after single-dose application of MI-883
Pharmacokinetic study of MI-883 in C57BL/6N mice (9–10 weeks old, body weight 20.8 to 24.5 g and average body weight across all groups 22.7 g, SD = 1.1 g) were performed following peroral (*p.o.*) and intraperitoneal (*i.p.*) administrations. Six-time points (10, 30, 120, 240, 480, and 1440 min) were set for this pharmacokinetic study, with each of the time point treatment groups included 4 animals (*n* = 4). There was also a control group of 4 animals per route. The levels of the parent compound MI-883 were determined by LC-MS/MS in the blood plasma, brain, kidney, liver, muscle (*Leg triceps*), bile, small intestine, colon, and adipose tissues over time after a single dose of 10 mg/kg. Mice were injected *i.p.* with 2,2,2-tribromoethanol at a dose of 150 mg/kg prior to drawing the blood. Blood collection was performed from the orbital sinus in BD microtainers tubes containing K2EDTA. Animals were sacrificed by cervical dislocation after the blood samples collection. After this, samples of brain, kidney, liver, muscle (*Leg triceps*), small intestine, colon, and adipose tissue were collected. Bile samples were taken from the gallbladder of mice. Urine and feces samples were collected using metabolic cages over the course of 1440 min (24 h) after compound administration. All samples were immediately processed, flash-frozen, and stored at −70 °C until subsequent analysis. Detection of the presence of potential metabolites (2-chloro-5-((4-(2-(4-chlorophenyl)-7-fluoroimidazo[1,2-a]pyridin-3-yl)-1H-1,2,3-triazol-1-yl)methyl)-N-methylbenzamide or 2-(4-chlorophenyl)-7-fluoro-3-(1H-1,2,3-triazol-4-yl)imidazo[1,2-*a*]pyridine) in plasma and tissues was performed. MI-883 was applied in 5%DMSO − 5%PEG300 − 90%corn oil formulation with a final concentration of the compound, 2 mg/ml. Animal treatment and sample preparation were conducted by the Animal Laboratory personnel at Enamine/Bienta.

Plasma samples (50 µl) were mixed with 200 µl of IS400 solution. After mixing by pipetting and centrifuging for 4 min at 6000 rpm, 2 µl of each supernatant was injected into the LC-MS/MS system. Solution of compound IS-19851 (400 ng/ml in acetonitrile-methanol mixture 1:1, v/v) was used as internal standard (IS400) for quantification of MI-883 in blood plasma, liver, kidney, muscle (*Leg triceps*), small intestine, and colon samples. Liver, kidney, muscle (*Leg triceps*), colon, and small intestine samples (weight 100 mg ±1 mg) were disintegrated in 500 µl of IS400 using zirconium oxide beads (120 mg ± 5 mg) in The Bullet Blender® homogenizer for 1–3 min at speed 8–12. After this, the samples were centrifuged for 3 min at 14,000 rpm, and 1 µl of each supernatant was injected into the LC-MS/MS system.

Brain and adipose tissue samples (weight 100 mg ±1 mg) were dispersed in 500 µl of IS400(80) using zirconium oxide beads (120 mg ± 5 mg) in The Bullet Blender® homogenizer for 30 s at speed 8. After this, the samples were centrifuged for 4 min at 14,000 rpm, and 1 µl of each supernatant was injected into the LC-MS/MS system. Solution of compound IS-19851 (400 ng/ml in water-acetonitrile-methanol mixture, 2:4:4, v/v/v) was used as internal standard

(IS400(80)) for quantification of MI-883 in the brain, adipose tissue, urine, and bile samples. Urine (cumulative) and bile samples were mixed with IS400(80) solution (1:20, v/v). After mixing by pipetting and centrifuging for 4 min at 6000 rpm, 0.5 µl of each supernatant was injected into the LC-MS/MS system. Cumulative feces samples (114 mg−249 mg) were transferred to BMT-20-S tubes with 4 stainless steel balls, mixed with 20 volumes of IS400(80) solution, and dispersed using ULTRA-TURRAX Tube Drive for 1 min at 3000 rpm within the 30 min thrice. After dispersing and centrifuging for 2 min at 14,000 rpm, 0.5 µl of each supernatant was injected into the LC-MS/MS system.

## MI-883 analysis following intraperitoneal and peroral administration

The concentration of MI-883 in plasma and tissues was determined using a high-performance liquid chromatography/tandem mass spectrometry (HPLC-MS/MS) method. Shimadzu HPLC system consisted of controller Prominence CBM20A2, isocratic pumps LC-10ADvp, an autosampler Prominence SIL-20AC, a sub-controller FCV-14AH, and a degasser DGU-14A. Mass spectrometric analysis was performed using the API 3000 (triple-quadrupole) instrument from AB Sciex (Canada) with an electro-spray (ESI) interface. The data acquisition and system control were performed using Analyst 1.5.2 software (AB Sciex, Canada). Calibration solutions and curves with final concentrations of 2500, 1000, 500, 250, 100, 50, 25, 10, 5, and 2 ng/ml were constructed using blank samples.

Pharmacokinetic calculations were performed as we described before[19].

Reagent, equipment and HPLC-MS/MS conditions are described in Supplementary Chapter 5.

## CYP enzymatic activity assays

Human recombinant CYP3A4 and CYP2B6 enzymes expressed from cDNA using a baculovirus-infected insect cells with human CYP450 reductase and cytochrome b5 in microsomal fraction (CYP450-Glo™ CYP3A4 Assay, CYP450-Glo™ CYP2B6 Assay, Promega, Hercules, CA) were used to evaluate interaction of MI-883 with these enzymes in vitro according to protocols we published before[19].

## mRNA sequencing using NGS (RNA-Seq) for transcriptome analysis

Total RNA from liver samples has been isolated using TRIZOL®. Total RNA was purified using Agencourt RNAClean XP Beads (Beckman Coulter, Inc.) and then the diluted RNA was quantified using Qubit 2.0 Fluorometer with Qubit RNA BR Assay Kit (Invitrogen) for precise measurement. The integrity of purified RNA was evaluated based on RIN (>7) acquired using the 2100 Bioanalyzer System with Total RNA Nano Chip (Agilent Technologies) according to the manufacturer's protocol.

The input amount of total RNA used in the libraries was 520 ng. Libraries were constructed with NEBNext Ultra II Directional RNA Library Prep Kit for Illumina (New England Biolabs Inc., Ipswich, MA, USA). The whole procedure was performed according to the protocol suggested by the manufacturer. The samples were multiplexed using suitable molecular barcodes by using the NEBNext Multiplex Oligos from Illumina. Adaptor enrichment was performed by using eight cycles of PCR.

The concentration of molecular libraries was measured using Qubit 2.0 Fluorometer with Qubit dsDNA HS Assay Kit (Invitrogen) and subsequently, the fragment sizes were analyzed by Bioanalyzer System 2100 using High Sensitivity DNA Chip (Agilent Technologies), with expected fragments between 200 and 600 bp in size. The molarity of individual libraries was calculated using determined concentrations and modal lengths. Each sample of the library was normalized to 14 nM concentration, and the normalized libraries were pooled and

sequenced on an Illumina NovaSeq6000 platform (Illumina) with paired-end sequencing of libraries with a read length of 150 b.

The quality of raw sequencing data was assessed using MultiQC tool. Fastq files were aligned to the mouse reference genome (mm10) using STAR aligner and featureCounts was used to make count matrices. Data was filtered to only include genes with a minimum 60 counts in all samples. Data were analyzed using the DESeq2 package of RStudio 2023.06.1 and R software v4.3.1 (https://www.R-project.org/). A pairwise comparison between MI-883-treated and control animals resulted in a list of differentially expressed RNAs ($P < 0.05$). P values were adjusted for multiple testing to reduce false discovery rate. For further analysis RNAs with more than twofold expression were selected (log2FC > 1 or < −1). Differential expression was analyzed using DESeq2 package (Fig. 4a). Volcano plot was visualized using R package ggplot2 and ggrepel. Gene ontology (GO) and pathway analyses were done with the R package clusterProfiler[64] using function enrichGO. $P$ value < 0.001 was defined as the significance threshold.

A heatmap was made using z-score normalized reads (z = [x−mean]/standard deviation) and visualized using the R package pheatmap. Hierarchical clustering was performed with the Ward method as a part of the pheatmap R package. The data presented in this publication have been deposited in NCBI's Gene Expression Omnibus database with GEO Series accession number GSE262786. https://www.ncbi.nlm.nih.gov/geo/query/acc.cgi?acc=GSE262786.

## *In*-silico molecular dynamics analysis

Receptor and Ligand Preparation: The crystal structure of the human wtCAR model was retrieved from the RCSB Protein Data Bank (www.rcsb.org, PDB code: 1XVP; resolution:2.6 Å). The PXR model was retrieved from a previous publication[23]. Ligands for docking were drawn using Maestro (v2022.4) and prepared using LigPrep to generate the three-dimensional conformation, adjust the protonation state to physiological pH (7.4), and calculate the partial atomic charges, with the force-field OPLS4. We employed a standard docking to accommodate MI-883 into CAR's and PXR's LBD using Glide[65,66]. Ligands were docked within a grid around 12 Å from the centroid of the crystallized ligand (CITCO) generating 10 poses per ligand. Next, four systems (wtCAR/CITCO, wtCAR/MI-883, PXR/SR12813, and PXR/MI-883) were prepared and minimized by adding hydrogens, adjusting the protonation states of amino acids, and fixing missing side-chain atoms and protein loops using Maestro PrepWizard (v2022.4). For each system, simulations of five 1 µs independent replicas were carried out. GraphPad Prism ver. 9.3.1. Software (GraphPad Software, Inc., San Diego, CA, United States) was used to perform statistical analysis.

Protein-ligand interactions were measured using the Simulation Event Analysis tool implemented in Maestro (Maestro v2022.4). The criteria for protein-ligand hydrogen bond is a distance of 2.5 Å between the donor and acceptor atoms (D − H···A); a donor angle of ≥120° between the donor-hydrogen-acceptor atoms (D − H···A); and an acceptor angle of ≥90° between the hydrogen-acceptor-bonded atom atoms (H···A − X). Protein-water or water-ligand hydrogen bond had a distance of 2.8 Å between the donor and acceptor atoms (D−H···A); a donor angle of ≥110° between the donor-hydrogen-acceptor atoms (D −H···A); and an acceptor angle of ≥90° between the hydrogen-acceptor-bonded atom atoms (H···A−X). Non-specific hydrophobic interactions are defined by hydrophobic sidechain within 3.6 Å of a ligand's aromatic or aliphatic carbons and π-π interactions required two aromatic groups stacked face-to-face or face-to-edge, within 4.5 Å of distance (distance definitions were taken from the Schrödinger software, Maestro v2022.4).

Distance calculations were carried out using Maestro event analysis tool (Schrödinger, LLC, New York, NY). Distances were calculated using the script *trj_asl_distance.py* selecting either the atom numbers of the residues (Cα) in the case of H2'−H6 distance or the center of mass of the Helices in the case of Helix 3−H12 distance as arguments.

Complementarily, to decipher the conformational dynamics of PXR-LBD induced by MI-883 binding (herein denoted as PXR-LBD/MI-883), we carried out 4.838 μs all-atom MD simulations for PXR/MI-883 and compared it against a classical agonist system (20 μs, PXR/SR12813)[23]. We utilized the co-crystal structure of PXR/SR12813 (PDB ID: 1NRL) as the starting configuration to model our antagonist system.

All molecular dynamics trajectories and raw data related to the protein-ligand interactions (as.dat and tsv files) within the simulations are available in the repository: codes: https://doi.org/10.5281/zenodo.10671315, https://doi.org/10.5281/zenodo.5772317 and https://doi.org/10.5281/zenodo.6355467.

## Cholesterol, C4, and lathosterol analysis

Cholesterol (Ch, total and free Ch), C4, and lathosterol determination were performed in liver tissue (25–80 mg fresh weight) using GC-MS according to Cohen et al.[67] (detailed protocol in Supplementary, Chapter 14). Internal standard (d7-cholesterol, Sigma Aldrich/Merck) was used. Solid phase extraction[68] and analysis were performed using LC-MS/MS as described previously[69]. Protein quantification was done using DC Protein Assay (Bio-Rad) according to manufacturer instructions (micro assay format). All samples were analyzed for protein content in triplicates and analyzed.

## Mass spectrometry analysis of lipids - lipidomics

Solvents, additives, lipid class internal standards, liver sample preparation, and UHPSFC/MS measurement are described in detail in Supplementary, Chapter 19. Lipid separation using UHPSFC (Acquity UPC$^2$ instrument from Waters; Milford, MA, USA) was performed on column Viridis BEH (100 × 3 mm, 1.7 μm) with the following conditions: the column temperature 60 °C, the flow rate 1.9 mL/min, and the injection volume 1 μL. The following linear gradient was performed using scCO$_2$ and methanol (30 mM ammonium acetate + 1% of water) used as a modifier: 0 min – 1% modifier, 1.5 min – 16% modifier, 4 min – 51% modifier, 7 min – 51% modifier, 7.51 min – 1% modifier, and the equilibration with the total run time of 8 min. The automatic back-pressure regulator (ABPR) was set to 1800 psi and the autosampler temperature to 4 °C. Methanol with 30 mM ammonium acetate and 1% of water was used as the make-up solvent with a flow rate of 0.25 mL/min. The SFC was connected with the hybrid quadrupole - time of flight (QTOF) mass spectrometer Synapt G2-Si (Waters; Milford, MA, USA) with following conditions: sensitivity mode applying positive ESI mode, the mass range of $m/z$ 150–1200, the capillary voltage of 3 kV, the sampling cone of 20 V, the source offset of 90 V, the source temperature of 150 °C, the desolvation temperature of 500 °C, the cone gas flow of 50 L/h, the desolvation gas flow of 1000 L/h, and the nebulizer gas flow of 4 bar. Mass spectra were acquired in the continuum mode with a scan time of 0.5 s and the peptide leucine enkephalin as the lock mass.

## Lipidomic data analysis and statistics

All UHPSFC/MS-QTOF spectra were acquired using MassLynx and processed as follows: noise reduction using the Compression tool followed by the lock mass correction and the conversion from continuum to centroid mode using the MassLynx tool (Accurate mass measure). Retention time windows of individual lipid classes were set to the methods created to get intensities (threshold of intensity 3000) using the MarkerLynx tool. Further, these methods for each lipid class were applied to a sequence of samples, which resulted in a sum table of all $m/z$ with corresponding intensity for each sample, exported as.txt file. and used for lipid identification, calculation of concentrations, and isotopic correction Type II. using LipidQuant Excel script[70]. All lipid intensities were normalized to respective lipid class internal standards and reported as concentrations (pmol/mg). Data were further analyzed using lipidr package[71]. PCA and log2 fold change values were calculated using log2 transformed and Pareto scaled data.

## Bile acid analysis in plasma, liver, stool, and bile

Materials and reagents and sample preparation have been described in our last paper[63] (see Supplementary, Chapter 12). Concentrations of bile acids in plasma, bile, and feces were determined using the UHPLC-MS method. The bile acids separation was conducted on the ACQUITY UPLC I-Class System (Waters, Milford, MA, USA) using a Triart C18 column (50 mm × 2.1 mm ID, 1.9 μm) from YMC, Japan. The column was maintained at 45 °C and was safeguarded by a disposable CrudCatcher in-line filter (Phenomenex, Torrance, CA, USA). The mobile phase flowed at a rate of 0.35 mL/min and consisted of solvent A (0.5 mmol/L ammonium acetate, 0.001% v/v acetic acid in water) and solvent B (0.5 mmol/L ammonium acetate, 0.001% v/v acetic acid in a 3:1 v/v mixture of methanol and acetonitrile). The gradient program followed this sequence. Initially, 40% (v/v) of solvent B was maintained for 0.2 min, after which there was a linear increase to 79% (v/v) from 0.2 to 9.0 min. This was succeeded by an isocratic washout with 95% (v/v) solvent B from 9.0 to 10.5 min, followed by equilibration with 40% (v/v) solvent B from 10.5 to 12 min. 2 μL of sample were injected into column. Column effluent was monitored with Xevo TQ-XS triple quadrupole detector operated in positive electrospray mode. Further details can be found in the publication by Uher et al.[72].

The processing of the UHPLC-MS data was performed using TargetLynx software (Waters, Milford, MA, USA). The ratio of an analyte's peak area and the internal standard was evaluated against a calibration curve, and the concentration for each bile acid was calculated. For further data analysis, the concentrations of individual BAs were summed to calculate the concentration of conjugated, unconjugated, 12-OH, non-12-OH, and total bile acids.

## The PXB-mouse Humanized liver mice experiment with MI-883

The use of animals for this study was approved by the Animal Ethics Committee of PhoenixBio (Resolution No.: 2554 approved on October 12, 2020). All the experimental procedures used to treat live animals in this study were approved by the Animal Ethics Committee of PhoenixBio.

PXB-mouse® are mice with background genotype: cDNA-uPAwild/+/SCID [cDNA-uPAwild/+: B6;129SvEv-Plau, SCID: C.B-17/Icr-scid /scid Jcl]. Mice contain human hepatocytes with an estimated replacement index of 70% or more in the liver, which is calculated based on the blood concentration of human albumin (h-Alb). Human hepatocytes for the experiment were obtained from BD195 (Corning Incorporated, Tewksbury, MA, USA).

The PXB mouse contains human hepatocytes and is utilized as a predictor of qualitative human hepatocytes/liver function[30].

Male mice ($n$ = 5 per group) were used for the experiments with the following characteristics: 19 to 23 weeks old on Day 0, body weights on Day 0 (19.3 to 22.3 g), and blood h-Alb Level (from 9.1 to 15.0 mg/mL).

Mice were fed ad libitum with 12 h-light/dark cycle (8:00–20:00). Mice group composition was randomized based on the arithmetic mean values for body weight and geometric mean values for blood h-Alb concentration. On the days of pre-dose blood sampling and/or administration, the observations were conducted prior to the pre-dose blood sampling and prior to the administration.

Individual body weights were taken once daily throughout the in-life phase. On the days of pre-dose blood sampling and/or administration, the body weights were taken prior to the pre-dose blood sampling and prior to the administration. The body weights were also taken prior to the fasting on Day 7.

MI-883 was applied in the following vehicle: corn oil (90%)/DMSO (5%)/PEG30 (5%) formulation of MI-883 2.5 mg/kg (1.5 mL) to treated animals. The same vehicle was used in the control group. MI-883 was applied via p.o. gavage using disposable plastic sondes (Fuchigami Kikai Co., Kyoto, Japan) at the dose of 2.5 mg/kg every day (in total seven doses -day 0-Day 7).

The target volume of blood (200 μL) was collected from all subject animals under isoflurane anesthesia via the retro-orbital plexus/

sinus using calibrated pipettes (Drummond Scientific Company, PA, USA) at each time point after 4 h of fasting. Isoflurane inhalation solution was also used at the terminal blood sampling. The livers were snap-frozen in two or more pieces in one tube in liquid nitrogen. These frozen liver samples were stored at −80 °C. Urine was directly collected from the urinary bladder using a syringe. Bile was directly collected from the gallbladder using a syringe.

Serum h-ALT1 concentration was determined based on an Enzyme-Linked ImmunoSorbent Assay (ELISA) developed by the Institute of Immunology Co., Ltd (Tokyo, Japan) (for details see Supplementary, Chapter 11).

### Plasma protein binding and plasma stability of MI-883
Plasma protein binding, and stability in human and mouse plasma have been tested as previously described[19].

### Metabolic stability in liver S9 fractions
Metabolic stability of MI-883 and CITCO in human liver S9 fraction (H0630.S9, XenoTech) and mouse liver S9 fraction (S9 from livers, pooled from male mouse (CD-1)) containing both phase I and phase II enzymes has been assessed as we described before[19]. To determine the phase II metabolism of MI-883, additional cofactors were used such as UDPGA (for glucuronidation), PAPS (for sulfation), and reduced glutathione (for conjugation of GSH). Shimadzu HPLC system including vacuum degasser, gradient pumps, reverse phase HPLC column, column oven, and autosampler with an API 4000 QTRAP mass spectrometer (Applied Biosystems/MDS Sciex (AB Sciex) with Turbo V ion source and TurboIonspray interface were used. The reference compound midazolam, which is metabolized both by phase I and phase II enzymes, was used. The elimination constant ($k_{el}$), half-life ($t_{1/2}$), and intrinsic clearance ($Cl_{int}$) were determined[19] (Table S1).

### Liver enzymatic TG measurement
The liver tissue (10 mg) was homogenized using a pre-cooled steel bead in 225 µL of methanol in TissueLyser II (30 Hz, 2 × 30 s) (Qiagen, Hilden, Germany). This was followed by the addition of 750 µL of ice-cold methyl tert-butyl ether. Samples were vortexed and incubated for 1 h on a rotating wheel. Next, 188 µL of LC/MS-grade water was added and, after vortexing, the mixture underwent a short incubation and centrifugation (14,000 × $g$, 4 °C, 3 min). The upper organic phase containing lipids was collected and 300 µL was transferred to a vial containing 40 µL of 10% Triton X-100 in methanol, followed by evaporation on a heating block with shaking function (300 rpm, 45 °C). LC/MS-grade water (100 µL) was added to the evaporated sample and the tube was vortexed (1000 rpm, 35 °C). The samples were measured using a TG assay (Beckman Coulter, Brea, CA, USA) and a glycerol assay (Randox laboratories, Crumlin, UK) on an AU480 biochemistry analyzer (Beckman Coulter, Brea, CA, USA). The glycerol value was subtracted from the TG value. Livers from seven ($n = 7$) vehicle-treated controls and ten ($n = 10$) MI-883-treated animals were used.

### Statistics and reproducibility
Histological and macroscopic analyses were performed with all liver samples; Fig. 6d presents a representative image of the analyses. Western blotting data were repeated with the same results, and representative data are shown in Fig. 6.

### Reporting summary
Further information on research design is available in the Nature Portfolio Reporting Summary linked to this article.

## Data availability
Data generated during the study is available in public repositories or at the request of the authors. All trajectories from the MD simulations, and raw data from geometrical and energy calculations are available on the Zenodo repository https://zenodo.org/uploads/10671315 with DOI: 10.5281/zenodo.10671315. The NGS RNA-Seq expression data generated in this study have been deposited in NCBI's Gene Expression Omnibus database and is accessible through GEO Series accession number GSE262786 https://www.ncbi.nlm.nih.gov/geo/query/acc.cgi?acc=GSE262786. Other data were deposited in *figshare* repository https://doi.org/10.6084/m9.figshare.25375909. Source data are provided with this paper.

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

## Acknowledgements

Funding Sources. The research was supported by the Technology Agency of the Czech Republic (The Council of the National Centre of Competence - TN02000109), the Ministry of Defence of the Czech Republic "Long Term Organization Development Plan 1011" – Healthcare Challenges of WMD II of the Military Faculty of Medicine Hradec Kralove, University of Defence, Czech Republic (project no: DZRO-FVZ22-ZHN II), the project"New Technologies for Translational Research in Pharmaceutical Sciences" /NETPHARM, project ID CZ.02.01.01/00/22_008/0004607 (co-funded by the European Union), and the Czech Science Foundation (GA ČR) No. 22-05167S (to P.P.). A.R., T.K., and P.P. acknowledge the European Union's Horizon 2020 research and innovation program under grant agreement No 825762, EDCMET project Metabolic effects of Endocrine Disrupting Chemicals: novel testing METhods and adverse outcome pathways (EDCMET). T.K. is funded by the fortune initiative and from TüCAD2 and CMIF. TüCAD2 and CMIF are funded by the Federal Ministry of Education and Research (BMBF) and the Baden-Württemberg Ministry of Science as part of the Excellence Strategy of the German Federal and State Governments. M.H. acknowledges the support of ERC Adv grant No. 101095860 sponsored by the European Research Council. The authors used services of the Czech Centre for Phenogenomics at the Institute of Molecular Genetics supported by the Czech Academy of Sciences RVO 68378050 and by the project LM2023036 Czech Centre for Phenogenomics provided by Ministry of Education, Youth and Sports of the Czech Republic (CZ.02.01.01/00/23_015/0008189 Upgrade of the large research infrastructure CCP III co-funded by EU and MEYS and CZ.02.1.01/0.0/0.0/18_046/0015861 CCP Infrastructure Upgrade II by MEYS and ESIF). The student grant SVV260 663 funds stipends for R.K. and M.K. The authors wish to acknowledge CSC – IT Center for Science, Finland, for the very generous computational resources and Prof. Antti Poso for their comments and support on resource acquisition. The CYP2B6-luc reporter plasmid (originally entitled B-1.6k/PB/XREM) was kindly donated by Dr. Hongbing Wang (University of Maryland School of Pharmacy, Baltimore, MD, USA), which we are grateful for. We thank Dr. Alena Mrkvicova for her help with the GEO database.

## Author contributions

J.D., T.S., M.K., I.P., O.B., and L.S. performed experiments; I.M., K, Š., J, H., and R.N. performed synthesis; K.D.,K.Ch., S.M. and P.P. performed animal experiments; A.R., T.K., J.S., D.P., R.vB., and R.K. performed in silico experiments and bioinformatics; M.Ch, M. H., M.L., M.H., and L.V. performed analyses; P.P. and R.N. designed the study; J.D., I.M., P.P., and R.N. wrote the paper; P.P., R.N., S.M., L.V., M.IS., O.B. and T.K. revised the manuscript.

## Competing interests

MI-883 belongs to a novel class of heterocyclic compounds protected by the family of patent applications (WO2020221380A1 PCT) with a priority date of 25 May 2020 in Europe, Canada, the United States, and Australia (EP3962915B1, CA3129981A1, US20220185822A1, and AU2020264679A1). The Australian patent was granted (AU2020264679B2) on December 8, 2022. The inventors of the patents are Petr Pavek, Radim Nencka, Jan Dusek, and Ivana Mejdrová. Applicants are the Charles University and the Institute of Organic Chemistry and Biochemistry, the Czech Academy of Sciences. Compound MI883 has been licensed to LipidEra Therapeutics B.V., Netherlands. The remaining authors declare no competing interests.

## Ethics

The research was conducted in accordance with journal policy. Co-authors were included if they met all authorship policies.

## Additional information

[1]Department of Pharmacology and Toxicology, Faculty of Pharmacy in Hradec Králové, Charles University, Hradec Králové, Czech Republic. [2]Institute of Organic Chemistry and Biochemistry, Czech Academy of Sciences, Prague, Czech Republic. [3]Czech Centre for Phenogenomics, Institute of Molecular Genetics of the Czech Academy of Sciences, Prague, Czech Republic. [4]First Faculty of Medicine, Charles University, Prague, Czech Republic. [5]Institute of Pharmacy, Pharmaceutical/Medicinal Chemistry and Tübingen Center for Academic Drug Discovery, Eberhard Karls University Tübingen, Tübingen, Germany. [6]Military Faculty of Medicine, University of Defence, Hradec Králové, Czech Republic. [7]Department of Analytical Chemistry, University of Pardubice, Faculty of Chemical Technology, Pardubice, Czech Republic. [8]Department of Biochemistry, Faculty of Medicine in Hradec Králové, Charles University, Hradec Králové, Czech Republic. [9]Institute of Medical Biochemistry and Laboratory Diagnostics, General University Hospital in Prague and First Faculty of Medicine, Charles University, Prague, Czech Republic. [10]Institute of Pharmacology, Faculty of Medicine in Hradec Králové, Charles University, Hradec Králové, Czech Republic. [11]PamGene, 's-Hertogenbosch, The Netherlands. [12]4th Department of Internal Medicine, General University Hospital in Prague and First Faculty of Medicine, Charles University, Prague, Czech Republic. [13]Section of Pharmacogenetics, Department of Physiology and Pharmacology, Karolinska Institutet, Stockholm, Sweden. [14]Dr. Margarete Fischer-Bosch-Institute of Clinical Pharmacology, Stuttgart, and University of Tuebingen, Tuebingen, Germany. [15]School of Pharmacy, Faculty of Health Sciences, University of Eastern Finland, Kuopio, Finland. [16]These authors contributed equally: Jan Dusek, Ivana Mejdrová. ✉e-mail: radim.nencka@uochb.cas.cz; pavek@faf.cuni.cz

