## [Transparent Peer Review file · Nature Communications]

The hypolipidemic effect of MI-883, the Combined CAR Agonist/ PXR Antagonist, in Diet-Induced Hypercholesterolemia Model

Corresponding Author: Professor Petr Pavek

Version 0:

Reviewer comments:

Reviewer #1

(Remarks to the Author)

In this paper, they developed and characterized a combined human CAR agonist/PXR antagonist MI-883. MI-883 was shown to regulate genes involved in xenobiotic metabolism and cholesterol/bile acid homeostasis. Lipidomics, bile acid metabolomics, and transcriptomics were performed in humanized PXR-CAR-CYP3A4/3A7 mice fed with high-fat and high-cholesterol diets. It was shown that MI-883 significantly reduced plasma cholesterol levels and enhanced fecal bile acid excretion. It is a significant study, and it is novel to me, it may lead to new treatments for metabolic disorders.

1. Biological half-life was investigated. But is MI-883 stable on the shelf? Soluble in water, or organic solvents?
2. Please justify the dosage of MI-883 to mice, and is that human relevant?
3. Lipidomics and bile acid profiling were performed, why not aqueous metabolomics? I understand it is about lipids and cholesterol, but they are connected, also long-chain fatty acids. If too much trouble to do the experiments, please justify and discuss about it.
4. Fecal excretion of BAs was increased, why? Any role of gut microbiome?

Reviewer #2

(Remarks to the Author)

The manuscript by Dusek et al. provides comprehensive characterization of a novel CAR agonist/PXR antagonist MI-883. This compound represents a first-in-class compound utilizing dual effect on two closely related nuclear receptors CAR and PXR. Both CAR and PXR have been shown to regulate intermediate metabolism including cholesterol metabolism. The relationship of these receptors is complex. While certain functions are regulated synergistically, some others response in opposite way to the activation of these NRs. PXR activation has been shown in both mice and humans to promote hypercholesterolemia. CAR activation instead appears to be beneficial in mice, however, the human data is missing due to lack of suitable ligands.

The study is broad and includes basic pharmacokinetic and toxicity studies. The pharmacodynamic properties are characterized in fairly detailed manner and effects on certain major metabolic endpoints are studied in preclinical models. The results suggest potential for a novel mechanism against hypercholesterolemia.

This study is important, thorough and opens new avenues towards therapeutic utilization of CAR/PXR. There are certain points that require further clarification or corrections.

Major points:

1. Sometimes the data is not very easy to read and interpret. Especially the experiments utilizing different reporter constructs need to be described more clearly. It is sometimes hard to follow which construct and what combination was used in each experiment. In the supplementary methods please clearly define the different experiment types: report gene assays, coactivator or corepressor recruitment and release, LBD assembly etc. Also, pay attention to the figures and figure legends so that the reader can easily understand what type of assay is used in each figure panel.
2. Is MI-883 a PXR antagonist or inverse agonist? In several experiments it appears to have a significant negative effect as

such.

3. What is the effect of MI-883 on intestinal NPC1L1 and FGF15?

4. PXB-study appears to be underpowered, and therefore the value of the results seems questionable. Did the authors calculate the samples size for this study?

5. Previous reports have suggested beneficial effects of CAR agonists on liver steatosis (PMC2773998). MI-883 seems, however, not to have any effect on liver fat. This should be discussed.

6. Although the results are promising it should be noticed as a limitation to the study that mouse is not a very good model for human hypercholesterolemia and therefore more evidence is needed in the future in sufficiently powered experiments with humanized animal models as well ultimately in human clinical studies.

Minor points

1. Sometime CAR is spelled Car even if the authors talk about protein.

2. In the introduction page 4 the authors state that "PXR activation elevates triglyceride and plasma LDL cholesterol (LDL-C) levels, triggers...". However, I don't think triglyceride level was increased in the study referred to. In fact, plasma triglyceride level is not usually increased by PXR activation. The authors also refer to the ref 12 stating that NPC1L1 was induced in intestine by PXR. However, Karpale et al could not detect intestinal NPC1L1 induction. However, such a finding has been reported in a different study PMC6413802.

3. In Fig 1j. please indicate clearly what the statistical significance stands for. Are all the values calculated against the cntr with empty vector? Or are some calculated against the corresponding contr?

4. Page 21 last sentence. The authors state "Overall, we determined that MI-883 decreases free cholesterol in the liver and increases fecal excretion of BAs but does not affect the plasma or liver contents of total cholesterol, triglycerides, or BAs." The statement that MI-883 does not affect plasma total cholesterol contradicts with results and the main message of the paper and therefore probably is a mistake.

5. Figure 6b, The volcano blot is quite difficult to read in its present form because of the two highly significant values. Could the authors cut the scale in the y-axis to enable better appreciation of the other results?

6. In Fig 6a and d, Con is not a good abbreviation to control as it could be interpreted to stand for concentration.

Reviewer #3

(Remarks to the Author)

In this manuscript, the authors describe a compound MI-883 which functions as a human CAR agonist and PXR antagonist/inverse agonist. Biochemical and cell-based assays (using cell lines or primary hepatocytes) have been used to evaluate the activity of MI-883 against PXR and CAR, and computational approach has been used to predict the mechanism of action of MI-883. The authors also use humanized PXR-CAR-CYP3A4/3A7 mice and the "PXB-mouse humanized liver mice carrying human hepatocytes" to demonstrate that MI-883 reduces cholesterol levels and enhances fecal bile acid excretion. Both the in vitro data on the dual action property (against PXR and CAR) and the in vivo data on the cholesterol-reducing effect of MI-883 are noteworthy results to the field. However, additional data or clarification will strengthen the conclusion on how MI-883 regulates PXR and CAR, and how the in vitro data on PXR/CAR function correlate with the in vivo data on cholesterol-lowering.

1. Have the authors used the humanized mice to evaluate the effect of MI-883 on PXR- and CAR- target genes? If so, do the data from mice follow the same trend as those observed in cell lines and primary hepatocytes?

2. Many target genes (for PXR and CAR activity) and biomarkers (for cholesterol reduction) have been described in data obtained from cell lines/hepatocytes and humanized mice. A table listing all these target genes and biomarkers affected by MI-883 in all experimental models (cell lines, hepatocytes, and mice) will help readers understand the correlation between these datasets.

3. Lines 164-165 "... Altogether, these data suggest that MI-883 exerts its inhibition of PXR by competitive antagonism (Fig. 1j, k).". The author also mentioned that MI-883 is an inverse agonist for PXR, which can't be explained by the proposed "competitive antagonism". How would the authors explain the inverse agonistic effect of MI-883 on PXR?

4. Lines 177-179 "...but does not suppress CYP2B6 activity (Supplementary Fig. S3d,e). We observed no other interactions with other tested nuclear receptors in reporter gene assays with MI-883 (Supplementary Fig. S4)". In these assays, MI-883 was tested for its agonistic effect. Did the authors also test whether MI-883 might be an antagonist for some of the NRs?

5. Lines 235-239 "(Supplementary information, Fig. S9). In cellular experiments with various two-hybrid assays, we confirmed that MI-883 did not recruit nuclear receptor coactivators NCOA1, NCOA2, and MED1 to PXR, but disrupted the interaction with corepressor NCOR2. Additionally, we showed that MI-883 blocked the recruitment of coactivator NCOA1 by agonist rifampicin, further confirming passive, competitive antagonism (Fig. 2o)." Why MI-883, as an antagonist/inverse agonist, disrupts PXR interaction with corepressor NCOR2?

6. This might be out of the scope of this study, but co-crystal structure of MI-883 with PXR and/or CAR will provide convincing mechanistic data on how MI-883 dually regulates PXR and CAR.

Version 1:

Reviewer comments:

Reviewer #1

(Remarks to the Author)

The revised paper addressed all my questions/concerns, I think it is ok to publish.

Reviewer #2

(Remarks to the Author)

The authors have done a good job in response to my comments. However, I still have some reservations related to the PXB-study discussion. I perfectly understand that there are financial restrictions, and it could be quite difficult to extend this study at the moment. While presenting the existing data is relevant, I don't think it is correct to say that "We did not detect a statistically significant effect on the total cholesterol concentration in the study due to a small sample size of animals (n=5)" (page 18). Although it is possible that with a larger sample size the authors could have been able to prove an effect on serum cholesterol parameters, also the opposite is possible. With the current results it is not possible to conclude this in one way or another. I would also think that it would be necessary to clearly state in the discussion that the current evidence on hypolipidemic effect of MI-883 is based on results in mouse and mRNA data in human cell models.

According to the mouse nomenclature (<https://www.informatics.jax.org/mgihome/nomen/gene.shtml#ps>) the protein symbols use all uppercase letters. The mouse and human proteins could be separated with h and m prefix. However, I leave this stylistic matter on journal's consideration.

Reviewer #3

(Remarks to the Author)

The authors have appropriately answered the questions raised at the last round of review.

REVIEWER COMMENTS

Reviewer #1 (Remarks to the Author):

In this paper, they developed and characterized a combined human CAR agonist/PXR antagonist MI-883. MI-883 was shown to regulate genes involved in xenobiotic metabolism and cholesterol/bile acid homeostasis. Lipidomics, bile acid metabolomics, and transcriptomics were performed in humanized PXR-CAR-CYP3A4/3A7 mice fed with high-fat and high-cholesterol diets. It was shown that MI-883 significantly reduced plasma cholesterol levels and enhanced fecal bile acid excretion. It is a significant study, and it is novel to me, it may lead to new treatments for metabolic disorders.

1. Biological half-life was investigated. But is MI-883 stable on the shelf? Soluble in water, or organic solvents?

Dear Reviewer,

Thank you for your comments. The metabolic stability data in human and mouse liver S9 fractions, along with the calculated biological half-lives ($t_{1/2}$), are presented in Table S1 of the Supplementary Information. The biological half-lives of MI-883 in plasma and liver tissues in mice are detailed in section 6.3 of the Supplementary Information, specifically in Table S2.

Our medicinal chemistry team analyzed the MI-883 sample that had been stored on the lab bench at room temperature for 6 months using HPLC/MS analysis and found no significant decomposition compared to the freshly prepared sample (see Figure R1).

A) Freshly prepared MI-883

B) MI-883 after standing under normal laboratory conditions with access to daylight for 6 months.

Figure R1. HPLC-MS analysis of samples of freshly prepared MI-883 (A) and a sample left for 6 months on a laboratory bench under daylight (B). Both samples were dissolved in HPLC methanol before analysis.

In addition, the compound is stable for at least two years in dimethyl sulfoxide (DMSO) as we confirm using published protocols ^{1,2}(see data in Table R1).

Table 1: LC/MS data for MI-883 stability in DMSO solution

time	μM	SD	RSD
0	1.012	0.028	2.776
2 year	1.007	0.030	2.986

Study protocol: Stability of MI-883 stock solution in DMSO after two years at -20°C
A 10 mM solution of MI-883 was prepared in DMSO and divided into multiple aliquots in 2 mL cryovials. The samples were stored at -20°C for two years in a standard laboratory freezer. Mass spectrometric analysis was conducted using LC/MS (Sciex QTRAP 7500). Stability was determined based on three independently prepared 1 μM solutions from stored stock.

The study results show that MI-883 is chemically stable under typical laboratory conditions in the DMSO stock solution at -20°C over two years. However, for critical pharmaceutical use, additional studies at different temperatures and under varied conditions (e.g., light exposure, humidity) may be necessary to fully assess long-term stability.

We observed no instability or decrease in activity when the compound was dissolved in DMSO for an extended period. Additionally, the compound demonstrated metabolic stability in both mouse and human plasma (see Table S1: Pharmacokinetic Parameters of MI-883 in the Supplementary Information).

MI-883 has low water solubility. We used several computational prediction tools to determine water solubility. The SwisADME tool (operated with two algorithms ^{3,4}) predicted a water solubility of 0.4 $\mu\text{g}/\text{mL}$ (0.9 μM) and 0.2 $\mu\text{g}/\text{mL}$ (0.4 μM). Using QikProp tool (Schrödinger software package), the QPlogS (predicted water solubility) is 0.017 μM . According to the ADMET Predictor software package (Simulation Plus), solubility in pure water (native solubility, S+Sw) was predicted to be 29 $\mu\text{g}/\text{mL}$. These predictions align with our observations and confirm the poor water solubility of MI-883, although we did not experimentally determine its solubility. The compound is well soluble in methanol and can dissolve in ethanol up to a concentration of 30 mM.

2. Please justify the dosage of MI-883 to mice, and is that human-relevant?

We used a dose of 5 mg/kg in our proof-of-concept studies. This dose was chosen based on several considerations:

1. In preliminary studies, we observed significant induction of Cyp2b10 mRNA in the liver of humanized PXR-CAR-CYP3A4/3A7 mice after a single dose of 5 mg/kg in oral gavage (with the same formulation). A single dose of 5-10 mg/kg is a common starting point based on scientific consensus and experimentation.
2. In a 28-day NOAEL/repeated dose oral toxicity study in rats (OECD Test Guideline for Chemicals No. 407), we observed no toxicity up to a dose of 10 mg/kg (Supplementary Information, Chapter 21).
3. Based on pharmacokinetic studies in male C57BL/6N mice (see Figure S3 in the Supplementary Information), we found that the concentration in the liver lysate after a single dose of 10 mg/kg was greater than 1000 ng/g for more than 18 hours (Figure S3 in the Supplementary Information, see Figure R2). This concentration corresponds to 2 μM . Considering the EC_{50} for CAR activation ($\text{EC}_{50}=0.38 \mu\text{M}$ in CAR LBD assembly assay) and the IC_{50}

for PXR antagonistic activity ($IC_{50}=2 \mu M$), MI-883 should have significant activity after a single dose of 10 mg/kg in mice.

Figure R2. Liver concentrations of MI-883 after a single 10 mg/kg dose applied via oral gavage. MI-883 has been applied via oral gavage (10 mg/kg) to male C57BL/6N mice. Liver samples were taken at indicated time intervals on dry ice before homogenization and processing. Each of the time points represents data from four animals. Samples (50 μ l) were mixed with an internal standard and injected into an LC-MS/MS system (for protocols, see section 6.2. in Supplementary Information).

However, after repeated administration, MI-883 is expected to accumulate in the body due to its significant volume of distribution (V_d) in mice ($V_d = 60$ mL calculated from data after a single dose application in mice). Therefore, the plasma concentration profile of MI-883 was simulated with a dosing regimen of 5 mg/kg every three days using a two-compartment model and pharmacokinetic parameters obtained from a single-dose study, analyzed with SAAM II pharmacokinetic software (see Figure R3). The simulation showed that plasma concentrations oscillate between 500-2000 ng/ml (1-4 μ M). Given that MI-883 reaches approximately twice the concentration in the liver compared to plasma (see Figure S3 in the Supplementary Information), it can be assumed that the dosing regimen (5 mg/kg over three days) produces sufficient hepatic concentrations to activate CAR and inhibit PXR without toxic effects in PXR-CAR-CYP3A4/3A7 mice.

Figure R3. Simulation of plasma MI-883 concentrations after the dose of 5 mg/kg per three days based on PK parameters obtained from the PK analysis after a single dose MI-883 application. SAAM II compartmental software (Epsilon) was used.

4. The dosage of MI-883 in the proof-of-concept study with a high-cholesterol diet was selected based on the same principles, set at 5 mg/kg every other day for 10 days.

This dose of 5 mg/kg is realistic when considering potential human dosing, as it should be divided by 12.3 when extrapolating and allometrically scaling the dose from animals to humans based on dose normalization to body surface area (see FDA guidelines Guidance for Industry Estimating the Maximum Safe Starting Dose in Initial Clinical Trials for Therapeutics in Adult Healthy Volunteers July 2005/04/28/2016).

3. Lipidomics and bile acid profiling were performed, why not aqueous metabolomics? I understand it is about lipids and cholesterol, but they are connected, also long-chain fatty acids. If too much trouble to do the experiments, please justify and discuss about it.

Based on your suggestion, we conducted metabolomic analyses of plasma samples from animals in the proof-of-concept study for the revised manuscript. The data are discussed in the manuscript and presented in Supplementary Information, Figure S18.

Principal component analysis (PCA) showed that MI-883 does not significantly affect the plasma metabolome (Figure R3A,B, Fig. S18). We only found that MI-883 significantly increased plasma levels of isoleucine (1.638-fold, $P = 0.0285$), 2-hydroxy-3-oxopropanoate (1.595-fold, $P = 0.0499$), and 5-aminopentanoic acid (also known as 5-aminovalerate) (1.375-fold, $P = 0.018$).

Isoleucine, which is proglucogenic and ketogenic, is one of the three branched-chain amino acids (BCAA) along with leucine and valine⁵. Mice undergoing fasting have elevated plasma levels of all three BCAAs, which are known as markers of fasting⁵. Additionally, blood levels of BCAA have long been associated with peripheral and hepatic insulin sensitivity primarily through activation of gluconeogenesis, but other molecular mechanisms have also been described^{6,7}. However, in our proof-of-concept study, we observed only a significant increase in plasma isoleucine (1.6-fold, see Figure R4), but not leucine and valine concentration (see Supplementary Data, Figure S18, Figure R3).

Endogenous 5-aminopentanoic acid is thought to be primarily a microbial metabolite of lysine catabolism produced by the gut microbiome or oral microflora, although it may also be produced endogenously (<https://hmdb.ca/metabolites/HMDB0003355>).

The prognostic value of elevated 2-hydroxy-3-oxopropanoate (1.595-fold, $P=0.0498$) is unknown from the literature. 2-hydroxy-3-oxopropanoate (syn. tartrate semialdehyde) is involved in ascorbate and aldarate metabolism and glyoxylate and dicarboxylate metabolism (KEGG PATHWAY database). The Human Metabolome Database (HMDB) reports that tartrate semialdehyde is a potential biomarker of consumption of certain foods, but this is irrelevant to our study.

Lipidomic analysis of long-chain fatty acids was conducted in the livers of PXR-CAR-CYP3A4/3A7 mice as part of a proof-of-concept study involving a high-fat diet. The results are presented in Figure 5j.

Figure R4. Metabolomics of plasma samples and the relative plasma levels of branched-chain amino acids (BCAAs) after treatment with MI-883 in the proof-of-concept studies. (A) Principal component analysis and (B) heatmap of metabolite concentration changes after MI-883 treatment. (C) Violin plots present data for specific metabolites, isoleucine, leucine, and valine, in the plasma of humanized PXR/CAR/CYP3A4/3A7 male mice treated with MI-883 in the proof-of-concept study with a high-fat diet. Relative levels of metabolites in plasma were determined using arbitrary units based on peak areas. $*P<0.05$.

4. Fecal excretion of BAs was increased, why? Any role of gut microbiome?

We suppose that fecal excretion was increased for several reasons, as outlined in the Discussion section (lines 634–645, page 31). The primary reasons include the upregulation of bile acid (BA) conjugation enzymes and efflux transporters in the liver, as well as the downregulation of the apical sodium-dependent bile acid transporter (Asbt) and organic solute α (Ost-alpha) transporters in the ileum, which regulate BA reabsorption and enterohepatic recirculation. As a result, BAs bypass reabsorption in the ileum and are excreted in the feces.

Our data and interpretation of MI-883's effects are consistent with findings published by Sberna *et al.* with the murine CAR ligand TCPOBOP.⁸ In our laboratory, we repeated this study. In wild-type animals treated with TCPOBOP, we similarly observed a significant increase in bile acid (BA) excretion into the stool, which was associated with the downregulation of certain BA uptake transporters in the ileum (our unpublished data).

PXR and CAR activation are known to limit pro-inflammatory microbes and BA-deconjugating microbes in the murine gut.

Microbes in CAR-null mice have been defined first, and the effect of CAR ablation on the BA deconjugating enzyme bile salt hydrolase (*bsh*) has been described⁹. In the recent paper by Dempsey *et al.*, it was shown that TCPOBOP reduces bacteria of two taxa in *Bifidobacterium spp.* corresponding to reduced gene abundance of *bsh* expressed in bacteria in the large intestinal content¹⁰. Decreased *bsh* activity may be connected with an increase in conjugated BA content and augmented elimination of conjugated BAs into feces due to decreased reabsorption. Therefore, the reviewer's comment is logical because bile salt dehydrogenase activity in the gut microbiome may further influence BA elimination and reabsorption after MI-883 treatment.

However, in the latter report, TCPOBOP did not significantly increase the colon's unconjugated and total BA pools (about 1.4-fold increase) in normal mice but significantly decreased the colon *bsh* activity (drop to 30% activity compared to control animals)¹⁰. Thus, reduced *bsh* activity may contribute but is unlikely to be critical for BA elimination into feces after treatment with TCPOBOP.

In our work we found a statistically significant increase of total bile acids in the stool (Fig. 5e) and a significant increase in unconjugated β MCA and CDCA in the stool after four-week treatment with MI-883 (Fig. 5f). We propose that the increased elimination of bile acids is primarily due to the downregulation of BA reabsorption in the ileum, and that the microbiome may play a role in modulating the BA metabolome. Unfortunately, we did not investigate the gut microbiome in this study. However, we have addressed this issue in more detail in the revised manuscript and included the reference by Dempsey *et al.* in the main body of the text (page 32, line 655).

Reviewer #2 (Remarks to the Author):

The manuscript by Dusek *et al.* provides comprehensive characterization of a novel CAR agonist/PXR antagonist MI-883. This compound represents a first-in-class compound utilizing dual effect on two closely related nuclear receptors CAR and PXR. Both CAR and PXR have been shown to regulate intermediate metabolism including cholesterol metabolism. The relationship of these receptors is complex. While certain functions are regulated synergistically, some others response in opposite way to the activation of these NRs. PXR activation has been shown in both mice and humans to promote cholesterolemia. CAR activation instead appears to be beneficial in mice, however, the human data is missing due to lack of suitable ligands.

The study is broad and includes basic pharmacokinetic and toxicity studies. The pharmacodynamic properties are characterized in fairly detailed manner and effects on certain major metabolic

endpoints are studied in preclinical models. The results suggest potential for a novel mechanism against hypercholesterolemia.

This study is important, thorough and opens new avenues towards therapeutic utilization of CAR/PXR. There are certain points that require further clarification or corrections.

Major points:

1. Sometimes the data is not very easy to read and interpret. Especially the experiments utilizing different reporter constructs need to be described more clearly. It is sometimes hard to follow which construct and what combination was used in each experiment. In the supplementary methods please clearly define the different experiment types: report gene assays, coactivator or corepressor recruitment and release, LBD assembly etc. Also, pay attention to the figures and figure legends so that the reader can easily understand what type of assay is used in each figure panel.

Dear Reviewer,

Thank you for your valuable comments. We have revised both the manuscript and the Supplementary Information (chapters 5.1 and 5.2) to improve clarity. Specifically, we have provided more detailed descriptions of the methods involving plasmids in both the manuscript and supplementary materials. Additionally, we have modified Figures 1 and 2, including their labels and legends, to ensure better clarity.

All DNA constructs and assays have been thoroughly described, with detailed information regarding their structure and origin, in the revised manuscript.

2. Is MI-883 a PXR antagonist or inverse agonist? In several experiments it appears to have a significant negative effect as such.

Thank you for the smart comment.

MI-883 exhibits both PXR antagonist and inverse agonist activities. We use the term "inverse agonist" as it is commonly used in pharmacological literature. An inverse agonist is a drug that, upon binding to a receptor, decreases the receptor's basal constitutive activity, demonstrating negative potency. Importantly, this reduction in basal activity is not due to the displacement of an endogenous agonist.¹¹ The PXR inverse agonism of compound MI-883 is evident in Figure 1i (reporter gene assay data) and Figures 1l and 1m (RT-qPCR data).

In contrast, MI-883 mediates the inhibition of rifampicin (an agonist)-induced activation of the luciferase reporter construct and rifampicin-induced CYP3A4 mRNA expression. This effect is referred to as PXR antagonism. By definition, antagonists do not produce an effect on their own but have an affinity for the receptor, preventing activation by an agonist¹¹. It has been shown that many antagonists can behave as inverse agonists (e.g., in the case of GPCR receptors).

This distinction is important when considering the existence of neutral PXR antagonists that do not exhibit inverse agonist activity (e.g., neutral PXR antagonists compound 89 and 97 in^{12,13}). This combination of activities (antagonistic/inverse agonistic) was also described and these definitions applied to other discovered potent PXR antagonists/inverse agonists such as SPA70 or compound 85^{13,14}.

Nevertheless, we accept your comment and concern. We understand that the classification might be confusing to some readers. Therefore, in the revised manuscript, we have refined the description of MI-883's inhibitory activities on PXR in section 2.5, opting to use the term "PXR antagonist" to encompass all its inhibitory functions.

3. What is the effect of MI-883 on intestinal NPC1L1 and FGF15?

The NPC1-like intracellular cholesterol transporter 1 (*Npc1l1*) plays a key role in intestinal cholesterol absorption. We found no significant effects of MI-883 on *Npc1l1* and *Fgf15* mRNA expression in the mouse ileum (see Figure R4). Moreover, the small heterodimer partner (*Shp*, *Nr0b2*), a sensitive target gene of FXR receptor controlling *Fgf15* expression, was not upregulated in the ileum after MI-883 treatment. These data are now included in the Supplementary Information (Figure S14) and are discussed in the main body of the revised manuscript. Also, the other authors did not find a significant effect of TCPOBOP, a murine *Car* ligand, on *Npc1l1* expression in the ileum of male mice¹⁵. Overall, these results suggest that MI-883 does not affect cholesterol absorption via *Npc1l1* in the intestine.

The *Abcg5/8* heterodimer is the primary neutral sterol transporter in the liver and intestine involved in hepatobiliary and transintestinal cholesterol excretion (TICE). Importantly, the complex mediates TICE, the pathway for fecal cholesterol elimination from plasma. In our proof-of-concept study with mice fed a high-fat diet (HFD), we observed no regulation of *Abcg5/Abcg8* mRNA in the intestine after treatment with MI-883 (Figure R5, Figure S14 in revised Supplementary Information). These results suggest that MI-883 does not significantly affect *Abcg5/Abcg8* expression and TICE in the gut.

Previous studies have shown that *Abcg5/8* mRNA is downregulated by TCPOBOP in both the liver and intestine^{8,16}. This observation is consistent with our unpublished experimental data with TCPOBOP, a highly potent murine *Car* agonist. However, MI-883, with its dual activity, did not downregulate *Abcg5/8* in either the liver or ileum in proof-of-concept studies with *PXR/CAR/CYP3A4/3A7* mice fed a high-fat diet.

Expression in ileum

Figure R5. Expression of *Npc1l1*, *Fgf15*, *Shp/Nr0b2*, *Abcg5*, and *Abcg8* mRNA in the ileum after treatment with MI-883 in humanized PXR-CAR-CYP3A4/3A7 mice fed with high-fat (HFD) or high-cholesterol (HCD) diets. Expression was analyzed employing RT-qPCR assays with TaqMan probes. The design of the proof-of-concept animal studies with HFD or HCD is described in Chapter 4.7.

4. PXB-study appears to be underpowered, and therefore the value of the results seems questionable. Did the authors calculate the samples size for this study?

We agree with the comment and plan to expand the study. We believe that this model is an appropriate model to test the effect of MI-883 because it expresses both human nuclear receptors PXR and CAR and also produces a human spectrum of lipoproteins and apolipoproteins, including VLDL lipoprotein, via human hepatocytes. However, the limitations of the humanized PXB liver mouse model are the cost of the model and the time required to generate animals by a commercial partner. It is not currently within our capabilities to extend the results with this model.

5. Previous reports have suggested beneficial effects of CAR agonists on liver steatosis (PMC2773998). MI-883 seems, however, not to have any effect on liver fat. This should be discussed.

Yes, the murine Car agonist TCPOBOP has been shown to ameliorate liver steatosis under nutritional stress in mice.

CAR activation was shown to alleviate hepatic steatosis *by down-regulation of lipogenic genes* such as the stearoyl-CoA desaturase-1 (*Scd1*), sterol regulatory element-binding protein 1c (*Srebp-1c*), and the acetyl-CoA carboxylase (*Acc/Acaca*) in mouse models subjected to metabolic/nutritional stress (high-fat diet), in leptin-deficient (*ob/ob*) or *Ldlr*^{-/-} mice, or mice on lithogenic diet^{8,16-18}.

On the other hand, under a normal diet (chow diet in mice), TCPOBOP mediates the up-regulation of lipogenic genes as well as neutral lipid (triglycerides and cholesterol esters) accumulation in the liver. This was associated with CAR-mediated up-regulation of fatty acid synthesis (*Fasn*), elongation (*Elovl6*), and esterification (*Gpat*) genes and the patatin-like phospholipase domain-containing protein 3 (*Pnpla3*) gene and its human orthologue through a Liver X receptor (LXR)-independent pathway^{19,20}. Another study showed the up-regulation of lipogenic genes by CAR in human hepatocytes likely through a non-classical S14 pathway²¹.

The effect of CAR activation on (non-alcoholic) metabolic dysfunction-associated steatohepatitis (NASH/MASH) pathogenesis in dietary-induced mouse models of MASLD was also reported with contradictory conclusions^{22,23}.

Recently, it was suggested based on experiments in CD-1 mice treated with TCPOBOP that the CAR activation recapitulates histological and gene expression changes characteristic of emerging steatotic liver disease (as analyzed using Red O staining), including secondary gene responses in liver nonparenchymal cells²⁰.

In our study, we did observe no significant accumulation of triglycerides in the livers of male mice after treatment with MI-883 for four weeks (5 mg/kg 3x per week) using both enzymatic and lipidomic approaches (Figure 5d and 5 i).

When we reanalyzed the expression of genes critical for liver steatosis and hepatic lipogenesis using RT-qPCR in 11 male animals (*n*=11), we found that fatty acid synthase (*Fasn*), involved mainly in palmitate synthesis, stearoyl-CoA desaturase (*Scd1*), the rate-limiting enzyme in the formation of monounsaturated fatty acids, and perilipin 2 (*Plin2*, involved in the formation of lipid droplets and an important protein in liver steatosis) display weak but statistically significant down-regulation in the livers of MI-883-treated humanized PXR-CAR-CYP3A4/3A7 male mice fed high-fat diet (see Figure R6 below). Consistently with the RT-qPCR data, *Scd1* mRNA has been significantly down-regulated by MI-883 in the livers (*n*=5) in NGS RNA Seq analysis ($\log_2FC_{DESeq} -0.4854$; $FDR_{DESeq} 0.0023$).

We therefore suggest that MI-883 has a weak or no significant effect on triglyceride synthesis in the proof-of-concept study involving humanized PXR-CAR-CYP3A4/3A7 mice treated with 5 mg/kg of MI-883 three times per week for four weeks. Additionally, we propose that CAR activation has complex and nutrition-dependent effects on liver steatosis and lipogenesis across various models.

These findings are discussed in the revised manuscript (page 27, lines 538-544) and are presented in the revised Supplementary Information as Figure S13.

Figure R6. Hepatic expression of important genes involved in lipid accumulation and triglyceride synthesis in humanized PXR-CAR-CYP3A4/3A7 mice fed with high-fat diet (HFD) and treated with MI-883 (5 mg/kg 3x per week) in the proof-of-concept study (n=11). The expression was analyzed employing RT-qPCR assays with TaqMan probes. The design of the proof-of-concept animal studies is described in Chapter 4.7. An unpaired parametric t-test was used to compare the two sets.

6. Although the results are promising it should be noticed as a limitation to the study that mouse is not a very good model for human hypercholesterolemia and therefore more evidence is needed in the future in sufficiently powered experiments with humanized animal models as well ultimately in human clinical studies.

Yes, we fully agree with this comment and acknowledge the limitations. We have addressed these limitations in our manuscript. Our future experiments will focus on extending the study using humanized live PXB mice (see Figure 7 and Figures S15 and S17).

To the best of our knowledge, this is currently the only available model that features humanized CAR and PXR expression in the liver, along with humanized cholesterol and lipoprotein homeostasis. However, this model is extremely expensive, labor-intensive, and provided by a commercial partner, which makes it challenging to scale up the study without the support of an investor.

Minor points

1. Sometime CAR is spelled Car even if the authors talk about protein.

We corrected the spelling in the whole article. Car stands for murine Car in the manuscript. Thank you.

2. In the introduction page 4 the authors state that “PXR activation elevates triglyceride and plasma LDL cholesterol (LDL-C) levels, triggers...”. However, I don’t think triglyceride level was increased in the study referred to. In fact, plasma triglyceride level is not usually increased by PXR activation. The authors also refer to the ref 12 stating that NPC1L1 was induced in intestine by PXR. However, Karpale et al could not detect intestinal NPC1L1 induction. However, such a finding has been reported in a different study PMC6413802.

Thank you for your valuable comments. Indeed, increased triglyceride (TG) plasma levels are not presented in the study by Karpale et al. (2021) in the British Journal of Pharmacology. They only found

a significant increase in plasma total fatty acids after rifampicin treatment. We have modified the relevant sentence in the manuscript according to your advice.

Regarding the comment on intestinal Npc1l1 expression, we have corrected the sentence and included a reference to the article published by Meng *et al.*²⁵ to specify that the induction of Npc1l1 was observed in PXR^{fl/fl} mice, but not in intestinal PXR knockout mice.

3. In Fig 1j. please indicate clearly what the statistical significance stands for. Are all the values calculated against the cntr with empty vector? Or are some calculated against the corresponding contr?

Thank you for your comment. Figure 1j has been edited for clarity. We also revised the statistical analysis to include ANOVA with multiple comparisons. Additionally, we indicate when wtPXR LBD is used as a control in the revised figure and the figure legend.

4. Page 21 last sentence. The authors state “Overall, we determined that MI-883 decreases free cholesterol in the liver and increases fecal excretion of BAs but does not affect the plasma or liver contents of total cholesterol, triglycerides, or BAs.” The statement that MI-883 does not affect plasma total cholesterol contradicts with results and the main message of the paper

Thank you for the comment. We corrected this sentence.

5: Figure 6b, The volcano blot is quite difficult to read in its present form because of the two highly significant values. Could the authors cut the scale in the y-axis to enable better appreciation of the other results?

Thank you for the comment. We have redone the volcano plot in Figure 6 according to your suggestion.

6. In Fig 6a and d, Con is not a good abbreviation to control as it could be interpreted to stand for concentration.

Thank you for the comment. We have made revisions.

Reviewer #3 (Remarks to the Author):

In this manuscript, the authors describe a compound MI-883 which functions as a human CAR agonist and PXR antagonist/inverse agonist. Biochemical and cell-based assays (using cell lines or primary hepatocytes) have been used to evaluate the activity of MI-883 against PXR and CAR, and computational approach has been used to predict the mechanism of action of MI-883. The authors also use humanized PXR-CAR-CYP3A4/3A7 mice and the “PXB-mouse humanized liver mice carrying human hepatocytes” to demonstrate that MI-883 reduces cholesterol levels and enhances fecal bile acid excretion. Both the in vitro data on the dual action property (against PXR and CAR) and the in vivo data on the cholesterol-reducing effect of MI-883 are noteworthy results to the field. However, additional data or clarification will strengthen the conclusion on how MI-883 regulates PXR and CAR, and how the in vitro data on PXR/CAR function correlate with the in vivo data on cholesterol-lowering.

1. Have the authors used the humanized mice to evaluate the effect of MI-883 on PXR- and CAR-target genes? If so, do the data from mice follow the same trend as those observed in cell lines and primary hepatocytes?

Yes, there is a strong correlation in the effects of MI-883 on many important genes involved in xenobiotic metabolism, cholesterol, and bile acid homeostasis across different models. The comparison of data from the various models used in this study, including both 2D and 3D primary human hepatocytes, is presented in Table R2 below.

In addition to the 2D and 3D primary human hepatocyte cultures, the supplementary information includes data obtained from mice with humanized PXB[®] livers containing human hepatocytes, with an estimated replacement index of 70% or more (Figures S15 and S17). These results show significant induction of CYP2B6 and UGT1A1 mRNA, while CYP3A4 mRNA levels have not significantly changed (Fig. S17). Thus, the data from the humanized liver mice are also consistent with findings from HepaRG cells (Figure 3b, Fig. S10) and primary human hepatocytes (Figures 3c, d, and h).

2. Many target genes (for PXR and CAR activity) and biomarkers (for cholesterol reduction) have been described in data obtained from cell lines/hepatocytes and humanized mice. A table listing all these target genes and biomarkers affected by MI-883 in all experimental models (cell lines, hepatocytes, and mice) will help readers understand the correlation between these datasets.

Thank you for the great suggestion. We have created the summary, and the data is presented in Table R2 below. However, we found that the clarity of the data presentation is compromised for several reasons.

Many genes show regulation without achieving statistical significance, and the context and experimental conditions vary between studies. Specifically, the data from humanized hPXR/hCAR/hCYP3A4/7 mice were obtained under nutritional stress (high-fat or high-cholesterol diet), whereas the cell models were not exposed to similar nutritional challenges. Additionally, some genes exhibit concentration-dependent regulation in HepaRG cells or time-dependent regulation in 3D spheroids of primary human hepatocytes. Moreover, MI-883 demonstrates both antagonistic and inverse agonistic PXR activities for certain genes. Finally, we observed mRNA down-regulation but protein up-regulation/stabilization for some genes. Furthermore, we utilized various methods to analyze transcriptome expression, including RT-qPCR and NGS RNA-Seq. Given these complexities, we anticipate that the table might be more informative in the Responses to Reviewers section but not in the text of the manuscript.

Table R2. Table listing important target genes regulated by MI-883 in various experimental models.

Gene:	Regulated by PXR/CAR based on data from HepaRG, HepaRG KO PXR and HepaRG KO CAR cells	Model				
		LS 174 T cell	HepaRG	Human hepatocytes	Humanized PXR-CAR-CYP3A mice	Humanized liver mice
CYP3A4/Cyp3a11	PXR>>CAR	Down-regulation ↓	Down-regulation ↓	Down-regulation (both 2D and 3D human hepatocytes) ↓	1.9-fold upregulation (n.s.)	No significant effect (P=0.421)
CYP2B6/Cyp2b10	CAR>PXR	n.d.	Significant induction ↑	Significant induction ↑	Significant induction ↑	Significant induction ↑
UGT1A1	PXR/CAR	n.d.	Significant induction ↑	Significant induction ↑	Significant induction ↑	Significant induction ↑
ABCC4/Abcc4	CAR>>PXR	n.d.	Significant induction ↑	Significant induction ↑	Significant induction ↑	Induction but no statistical significance (P=0.341) ↑
LDLR	PXR	n.d.	Significant up-regulation ↑	Up-regulation in 2D PHH ↑ Up-regulation in 2D PHH ↑ by SPA70	∅	∅, trend for ↑ (P=0.46)
SREBF1	CAR	n.d.	Significant down-regulation ↓	∅ in 2D, trend for ↓ Significant down-regulation ↓ in 3D PHH*	Significant down-regulation of mRNA (RT-qPCR and NGS RNA-Seq). ↓ A slight decrease of the mature protein ↓	Decrease, but no statistical significance (P=0.548) ↓
INSIG1	PXR/CAR	n.d.	Significant up-regulation ↑	weak ↑ ↑ by SPA70	Up-regulation of protein ↑	n.d.
HMGCR		n.d.	∅, trend for ↑	∅ in 2D PHH;	No significant effect on protein ∅, significant mRNA upregulation (1.3-fold) ↑	No significant effect ∅
HMGCSS2		n.d.	Down-regulation, but no statistical significance ↓	∅, trend for ↑	∅	Significant induction (P=0.016) ↑
CYP7A1	PXR, CAR?	n.d.	Significant down-regulation ↓	Significant down-regulation ↓	Significant down-regulation of mRNA ↓, but upregulation of protein ↑	n.d.
SQLE		n.d.	Significant down-regulation ↓ of rifampicin effect	∅, trend for ↑	Significant induction ↑	n.d.
PCSK9		n.d.	∅, trend for ↓	∅, trend for ↑	∅	n.d.
SCD1/Scd1	CAR	n.d.			Significant down-regulation of mRNA ↓	n.d.
Fasn	CAR	n.d.	Significant down-regulation ↓	Significant down-regulation ↓ in 3D PHH*	Significant down-regulation of mRNA ↓	n.d..

∅ – no effect on expression; ↓↓- down-regulation and statistically significant down-regulation (bold); ↑↑-up-regulation and statistically significant up-regulation (bold); * data not presented; PHH -primary human hepatocytes, n.d.-not determined; n.s. -not statistically significant.

3. Lines 164-165 "... Altogether, these data suggest that MI-883 exerts its inhibition of PXR by competitive antagonism (Fig. 1j, k).". The author also mentioned that MI-883 is an inverse agonist for PXR, which can't be explained by the proposed "competitive antagonism". How would the authors explain the inverse agonistic effect of MI-883 on PXR?

Thank you for the insightful question and comment. We have corrected the sentence to read: "Altogether, these data suggest that MI-883 exerts PXR inhibition through interaction with ligand-binding pocket (LBP) (Figure 1j,k)."

We hypothesize that the endogenous (constitutive) PXR activity may be influenced in part by endogenous ligands or components of the culture medium. However, there may also be other mechanisms, such as the inhibition of PXR nuclear translocation or post-translational modifications of PXR in the presence of a PXR antagonist. These mechanistic aspects are not yet well studied and understood.

4. Lines 177-179 "...but does not suppress CYP2B6 activity (Supplementary Fig. S3d,e). We observed no other interactions with other tested nuclear receptors in reporter gene assays with MI-883 (Supplementary Fig. S4)". In these assays, MI-883 was tested for its agonistic effect. Did the authors also test whether MI-883 might be an antagonist for some of the NRs?

Thank you for the idea. Yes, we conducted luciferase reporter assays in antagonistic "mode" for several nuclear receptors using their prototype ligands in the revised manuscript (Figure R7). MI-883 at a concentration of 10 μ M appears to be a weak inhibitor of the estrogen receptor alpha (ER α) but does not inhibit the other nuclear receptors. This information is included in the Supplementary Information.

Figure R7. Inhibition of farnesoid X (FXR), thyroid receptor- α (TR α), aryl hydrocarbon receptor (AhR), peroxisome proliferator-activated receptors (PPAR) α and γ , glucocorticoid receptor α (GR α), liver X receptor α (LXR α) and estradiol receptor- α (ER α) in luciferase reporter gene assays performed in transiently transfected HepG2 cells (the same protocol as for Figure S4). Treatment with 5 and 10 μ M MI-883 has been performed together with ligands of the nuclear receptors and AhR transcription factor: 1 μ M 6 α -ethyl-chenodeoxycholic acid (6-ECDCA), 10 μ M L-thyroxin, 10 μ M 3-methylcholanthrene (an

*agonist of the AhR transcription factor), 10 μM rosiglitazone (PPARα), 10 μM GW7647 (PPARα), 100 nM dexamethasone, 10 μM GW3965 (LXRα agonist), and 10 μM estradiol. Data are presented as a fold activation to control (vehicle DMSO 0.1%-treated) cells (set to be 0%). Activities of the ligands are set to be 100%. ANOVA with post-hoc test was used for statistical analysis. *P<0.05, statistically significant inhibition of ERα activity.*

5. Lines 235-239 “(Supplementary information, Fig. S9). In cellular experiments with various two-hybrid assays, we confirmed that MI-883 did not recruit nuclear receptor coactivators NCOA1, NCOA2, and MED1 to PXR, but disrupted the interaction with corepressor NCOR2. Additionally, we showed that MI-883 blocked the recruitment of coactivator NCOA1 by agonist rifampicin, further confirming passive, competitive antagonism (Fig. 2o).” Why MI-883, as an antagonist/inverse agonist, disrupts PXR interaction with corepressor NCOR2?

This is an excellent question, but it is challenging to answer. Currently, there is limited information regarding the interaction of PXR with NCOR1 and NCOR2/SMRT in the presence of a PXR antagonist. Data involving rifampicin, a PXR agonist, suggest an unexpected stimulation of the PXR-LBD/SMRT-VP16 interaction following rifampicin treatment²⁶. Thus, the interactions of PXR with co-repressors remain largely unexplored.

Unfortunately, for the manuscript, we were unable to perform the MARCoNI assay with recombinant PXR-LBD, as we did with CAR-LBD (Figure 2m).

6. This might be out of the scope of this study, but co-crystal structure of MI-883 with PXR and/or CAR will provide convincing mechanistic data on how MI-883 dually regulates PXR and CAR.

We have been struggling with the PXR-MI-883 co-crystal for an extended period and have created many amorphous particles in our attempts. We will continue our experiments. For the manuscript, we utilized the molecular dynamics simulation model from our previous publication²⁷. We agree that the crystals are crucial for future modification of the MI-883 structure and for our understanding of the nature of PXR antagonism/inverse agonism by MI-883. Therefore, we will continue our efforts to develop co-crystals, building on the recent work by Garcia-Maldonado et al.²⁸.

Petr Pavek and Radim Nencka

10 November 2024

References:

- 1 Kozikowski, B. A. *et al.* The effect of freeze/thaw cycles on the stability of compounds in DMSO. *J Biomol Screen* **8**, 210-215 (2003). <https://doi.org/10.1177/1087057103252618>
- 2 Engeloch, C. *et al.* Stability of screening compounds in wet DMSO. *J Biomol Screen* **13**, 999-1006 (2008). <https://doi.org/10.1177/1087057108326536>
- 3 Delaney, J. S. ESOL: estimating aqueous solubility directly from molecular structure. *J Chem Inf Comput Sci* **44**, 1000-1005 (2004). <https://doi.org/10.1021/ci034243x>
- 4 Ali, J., Camilleri, P., Brown, M. B., Hutt, A. J. & Kirton, S. B. Revisiting the general solubility equation: in silico prediction of aqueous solubility incorporating the effect of topographical polar surface area. *Journal of chemical information and modeling* **52**, 420-428 (2012). <https://doi.org/10.1021/ci200387c>
- 5 Canfield, C. A. & Bradshaw, P. C. Amino acids in the regulation of aging and aging-related diseases. *Translational Medicine of Aging* **3**, 70-89 (2019).
- 6 Lynch, C. J. & Adams, S. H. Branched-chain amino acids in metabolic signalling and insulin resistance. *Nat Rev Endocrinol* **10**, 723-736 (2014). <https://doi.org/10.1038/nrendo.2014.171>
- 7 Goffredo, M. *et al.* A Branched-Chain Amino Acid-Related Metabolic Signature Characterizes Obese Adolescents with Non-Alcoholic Fatty Liver Disease. *Nutrients* **9** (2017). <https://doi.org/10.3390/nu9070642>
- 8 Sberna, A. L. *et al.* Constitutive androstane receptor activation stimulates faecal bile acid excretion and reverse cholesterol transport in mice. *J Hepatol* **55**, 154-161 (2011). <https://doi.org/10.1016/j.jhep.2010.10.029>
- 9 Little, M. *et al.* Understanding the physiological functions of the host xenobiotic-sensing nuclear receptors PXR and CAR on the gut microbiome using genetically modified mice. *Acta Pharm Sin B* **12**, 801-820 (2022). <https://doi.org/10.1016/j.apsb.2021.07.022>
- 10 Dempsey, J. L. *et al.* Pharmacological Activation of PXR and CAR Downregulates Distinct Bile Acid-Metabolizing Intestinal Bacteria and Alters Bile Acid Homeostasis. *Toxicol Sci* **168**, 40-60 (2019). <https://doi.org/10.1093/toxsci/kfy271>
- 11 Parra, S. & Bond, R. A. Inverse agonism: from curiosity to accepted dogma, but is it clinically relevant? *Curr Opin Pharmacol* **7**, 146-150 (2007). <https://doi.org/10.1016/j.coph.2006.10.005>
- 12 Li, Y. *et al.* Building a Chemical Toolbox for Human Pregnane X Receptor Research: Discovery of Agonists, Inverse Agonists, and Antagonists Among Analogs Based on the Unique Chemical Scaffold of SPA70. *J Med Chem* **64**, 1733-1761 (2021). <https://doi.org/10.1021/acs.jmedchem.0c02201>
- 13 Li, Y. *et al.* Design and Optimization of 1H-1,2,3-Triazole-4-carboxamides as Novel, Potent, and Selective Inverse Agonists and Antagonists of PXR. *J Med Chem* **65**, 16829-16859 (2022). <https://doi.org/10.1021/acs.jmedchem.2c01640>
- 14 Lin, W. *et al.* SPA70 is a potent antagonist of human pregnane X receptor. *Nat Commun* **8**, 741 (2017). <https://doi.org/10.1038/s41467-017-00780-5>
- 15 Lickteig, A. J., Csanaky, I. L., Pratt-Hyatt, M. & Klaassen, C. D. Activation of Constitutive Androstane Receptor (CAR) in Mice Results in Maintained Biliary Excretion of Bile Acids Despite a Marked Decrease of Bile Acids in Liver. *Toxicol Sci* **151**, 403-418 (2016). <https://doi.org/10.1093/toxsci/kfw054>
- 16 Cheng, S. *et al.* Activation of Constitutive Androstane Receptor Prevents Cholesterol Gallstone Formation. *Am J Pathol* **187**, 808-818 (2017). <https://doi.org/10.1016/j.ajpath.2016.12.013>
- 17 Dong, B. *et al.* Activation of nuclear receptor CAR ameliorates diabetes and fatty liver disease. *Proc Natl Acad Sci U S A* **106**, 18831-18836 (2009). <https://doi.org/10.1073/pnas.0909731106>

- 18 Gao, J., He, J., Zhai, Y., Wada, T. & Xie, W. The constitutive androstane receptor is an anti-obesity nuclear receptor that improves insulin sensitivity. *The Journal of biological chemistry* **284**, 25984-25992 (2009). <https://doi.org/10.1074/jbc.M109.016808>
- 19 Marmugi, A. *et al.* Activation of the Constitutive Androstane Receptor induces hepatic lipogenesis and regulates Pnpla3 gene expression in a LXR-independent way. *Toxicol Appl Pharmacol* **303**, 90-100 (2016). <https://doi.org/10.1016/j.taap.2016.05.006>
- 20 Sonkar, R., Ma, H. & Waxman, D. J. Steatotic liver disease induced by TCPOBOP-activated hepatic constitutive androstane receptor: primary and secondary gene responses with links to disease progression. *Toxicol Sci* **200**, 324-345 (2024). <https://doi.org/10.1093/toxsci/kfae057>
- 21 Breuker, C. *et al.* Hepatic expression of thyroid hormone-responsive spot 14 protein is regulated by constitutive androstane receptor (NR1I3). *Endocrinology* **151**, 1653-1661 (2010). <https://doi.org/10.1210/en.2009-1435>
- 22 Baskin-Bey, E. S., Anan, A., Isomoto, H., Bronk, S. F. & Gores, G. J. Constitutive androstane receptor agonist, TCPOBOP, attenuates steatohepatitis in the methionine choline-deficient diet-fed mouse. *World J Gastroenterol* **13**, 5635-5641 (2007).
- 23 Yamazaki, Y., Moore, R. & Negishi, M. Nuclear receptor CAR (NR1I3) is essential for DDC-induced liver injury and oval cell proliferation in mouse liver. *Lab Invest* **91**, 1624-1633 (2011). <https://doi.org/10.1038/labinvest.2011.115>
- 24 Karpale, M., Hukkanen, J. & Hakkola, J. Nuclear Receptor PXR in Drug-Induced Hypercholesterolemia. *Cells* **11** (2022). <https://doi.org/10.3390/cells11030313>
- 25 Meng, Z. *et al.* The atypical antipsychotic quetiapine induces hyperlipidemia by activating intestinal PXR signaling. *JCI Insight* **4** (2019). <https://doi.org/10.1172/jci.insight.125657>
- 26 Takeshita, A., Taguchi, M., Koibuchi, N. & Ozawa, Y. Putative role of the orphan nuclear receptor SXR (steroid and xenobiotic receptor) in the mechanism of CYP3A4 inhibition by xenobiotics. *The Journal of biological chemistry* **277**, 32453-32458 (2002). <https://doi.org/10.1074/jbc.M111245200>
- 27 Rashidian, A. *et al.* Discrepancy in interactions and conformational dynamics of pregnane X receptor (PXR) bound to an agonist and a novel competitive antagonist. *Comput Struct Biotechnol J* **20**, 3004-3018 (2022). <https://doi.org/10.1016/j.csbj.2022.06.020>
- 28 Garcia-Maldonado, E. *et al.* Chemical manipulation of an activation/inhibition switch in the nuclear receptor PXR. *Nat Commun* **15**, 4054 (2024). <https://doi.org/10.1038/s41467-024-48472-1>

REPLY TO REVIEWER COMMENTS

Final revisions for Nature Communications manuscript NCOMMS-24-14659A

REVIEWERS' COMMENTS

Reviewer #1 (Remarks to the Author):

The revised paper addressed all my questions/concerns, I think it is ok to publish.

Reviewer #2 (Remarks to the Author):

The authors have done a good job in response to my comments. However, I still have some reservations related to the PXB-study discussion. I perfectly understand that there are financial restrictions, and it could be quite difficult to extend this study at the moment. While presenting the existing data is relevant, I don't think it is correct to say that "We did not detect a statistically significant effect on the total cholesterol concentration in the study due to a small sample size of animals (n=5)" (page 18). Although it is possible that with a larger sample size the authors could have been able to prove an effect on serum cholesterol parameters, also the opposite is possible. With the current results it is not possible to conclude this in one way or another. I would also think that it would be necessary to clearly state in the discussion that the current evidence on hypolipidemic effect of MI-883 is based on results in mouse and mRNA data in human cell models.

According to the mouse nomenclature

(<https://www.informatics.jax.org/mgihome/nomen/gene.shtml#ps>) the protein symbols use all uppercase letters. The mouse and human proteins could be separated with h and m prefix. However, I leave this stylistic matter on journal's consideration.

Reviewer #3 (Remarks to the Author):

The authors have appropriately answered the questions raised at the last round of review.

Dear Reviewers,

Thank you. We are glad that you like the manuscript. We are grateful for your help.

We thank reviewer#2 for the valuable comment. We modified the chapter in the Results section. We also stress that the effect of MI-883 is based mainly on the results in humanized PXR-CAR-CYP3A mice and human hepatocyte models.

We carefully reconsidered the nomenclature for protein and gene symbols in mice according to the journal guidelines. Gene names and gene mRNA are indicated in italics. Human, but not mouse, protein symbols use uppercase letters.

Sincerely,

Petr Pavek and Radim Nencka
On behalf of all co-authors